# UNI-COT: TOWARDS UNIFIED CHAIN-OF-THOUGHT REASONING ACROSS TEXT AND VISION

**Luozheng Qin**[1,*] **Jia Gong**[1,*†] **Yuqing Sun**[1,*] **Tianjiao Li**[3] **Haoyu Pan**[1]
**Mengping Yang**[1] **Xiaomeng Yang**[1] **Chao Qu**[2] **Zhiyu Tan**[2,1,†‡] **Hao Li**[2,1†]
[1]Shanghai Academy of AI for Science  [2]Fudan University  [3]Nanyang Technological University
{qinluozheng, gongjia, sunyuqing}@sais.org.cn

## ABSTRACT

Chain-of-Thought (CoT) reasoning has proven effective in enhancing Large Language Models (LLMs) on complex tasks by decomposing problems into step-wise solutions. However, extending CoT to multi-modal settings remains challenging, as it requires modeling transitions of visual states alongside textual reasoning. Existing approaches often underperform due to limited capacity to model visual transitions or fragmented architectures. To overcome this limitation, we introduce Uni-CoT, a Unified Chain-of-Thought framework that captures structured visual transitions and seamlessly aligns them with textual logic, enabling coherent multimodal reasoning. To mitigate the computational and training challenges inherent to multi-modal reasoning, Uni-CoT introduces a two-level reasoning paradigm: a macro-level CoT for high-level planning and a micro-level CoT for localized subtask execution. This hierarchical design reduces computational overhead while maintaining coherence. Additionally, Uni-CoT incorporates a structured training paradigm with auxiliary tasks to stabilize optimization and improve generalization. Experiments on reasoning-driven image generation and understanding benchmarks demonstrate that Uni-CoT achieves state-of-the-art performance and remarkable generalization, underscoring its potential for complex multi-modal reasoning. **Code: https://github.com/Fr0zenCrane/UniCoT**.

## 1 INTRODUCTION

Chain-of-Thought (CoT) (Wei et al., 2022) enables Large Language Models to solve complex problems by generating explicit intermediate steps before producing the final answer. Motivated by its success (Feng et al., 2020; Lu et al.), recent works (Zhang et al.; Zheng et al., 2023) have paid efforts to extend CoT to multi-modal domain, aiming to endow Multi-modal LLMs (MLLMs) with analogous reasoning abilities for challenging vision–language tasks (Xu et al., 2024; Mu et al., 2023).

Despite recent progress, modeling coherent multi-modal reasoning remains challenging. Zhang et al. (2025a); Yang et al. (2025b); Huang et al. (2025) explored to strengthen the text-based reasoning of MLLMs via reinforcement learning, but performed poorly on visually grounded tasks such as geometry or navigation (Wu et al., 2024b; Cheng et al., 2025). Humans, by contrast, can solve this problem easily by integrating visual state transitions into reasoning, e.g., updating a map during navigation. This gap calls for models that explicitly embed visual transitions rather than approximating them through text. Programmatic operations such as cropping, drawing, or plotting (Gupta & Kembhavi, 2023; Surís et al., 2023; Hu et al., 2024a) approximate local transitions but fail to capture global structural changes critical for tasks like jigsaw solving. Other approaches couple MLLMs with visual generators to model large visual change (Guo et al., 2025b; Jiang et al., 2025), but their loose integration often leads to fragmented reasoning and incoherent transitions.

To address these challenges, we propose Uni-CoT, a Unified Chain-of-Thought framework that integrates *structural visual transitions* with *coherent text reasoning* as shown in Figure 1. Uni-CoT

---

*Equal contribution
†Corresponding author
‡Project leader

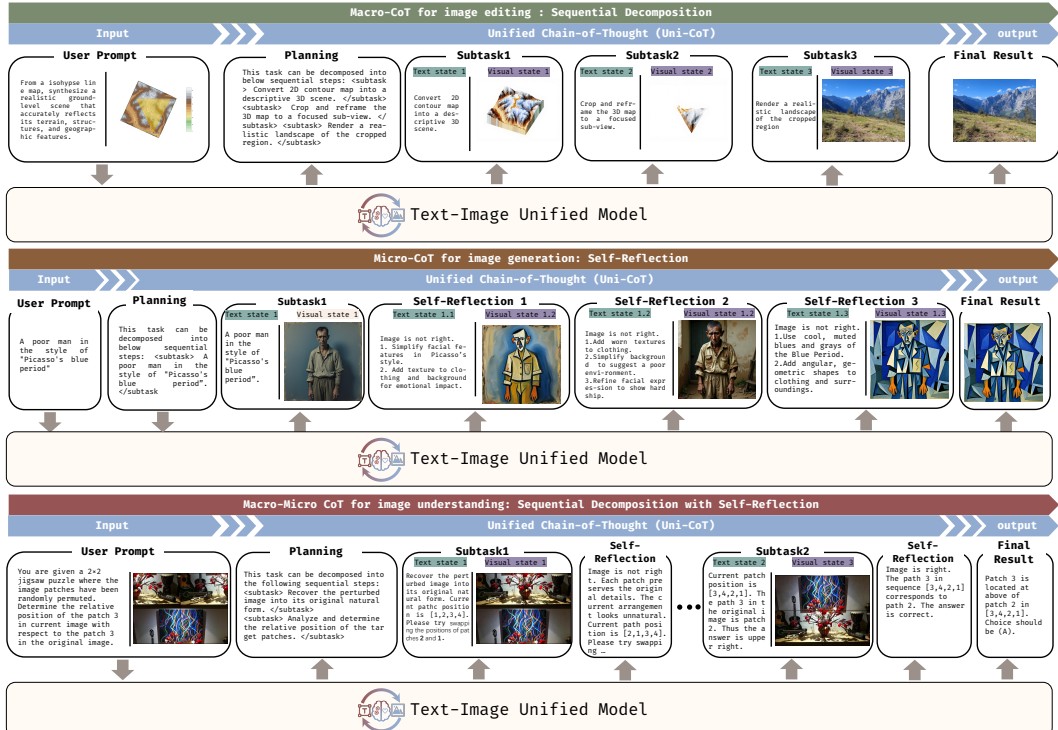

Figure 1: The multi-modal reasoning trajectory of Uni-CoT. Uni-CoT extends Chain-of-Thought to the multi-modal domain, enabling a unified model to conduct step-by-step reasoning across text and images, thereby supporting complex tasks in both image generation and understanding.

is built on a unified model (Deng et al., 2025) capable of both image understanding and generation, reducing discrepancies between reasoning and visual dynamics and enabling end-to-end training for coherent multi-modal reasoning. Nevertheless, realizing unified reasoning remains difficult due to two factors: (1) Computational burden: Multi-modal reasoning requires generating both textual and visual intermediates, inflating token lengths by orders of magnitude over text-only reasoning and resulting in substantial computational overhead (see Section 2). (2) Training instability: Long sequences complicates long-range dependency modeling and optimization, while interleaved image–text generation further destabilizes training due to mismatched cross-modal dynamics.

Inspired by human cognition, where complex problems are solved through hierarchical organization of reasoning (Broadbent, 1977; Newell, 1994), we propose a macro–micro hierarchical CoT framework for multi-modal reasoning to reduce complexity. At the macro level, the model first sketches a global strategy, decomposes the task into manageable subtasks, and then integrates their results into a coherent solution while abstracting away execution details. At the micro level, the model focuses on solving each subtask in isolation, filtering out irrelevant context. To further improve efficiency, the micro CoT is formulated as a Markov Decision Process (MDP), constraining dependencies to local states. This macro–micro restructuring converts long, entangled reasoning trajectories into modular blocks, substantially reducing redundant token interactions in the attention layers and lowering the complexity of multi-modal reasoning from quadratic to nearly linear.

Moreover, we decompose multi-modal CoT learning into two parts to achieve stable and effective training: (1) Macro-Level CoT Modeling: model is refined on interleaved text–image content to acquire global planning and final synthesis; (2) Micro-Level CoT Modeling: subtask execution is cast as an MDP-style process, augmented with four auxiliary tasks (e.g., action generation, reward estimation) to facilitate efficient learning. This decoupled paradigm provides supervision at both global and local levels, enabling scalable training for complex multi-modal reasoning tasks.

With these designs, Uni-CoT trains efficiently on 8×A100 GPUs and achieves state-of-the-art results on image generation (Niu et al., 2025; Ghosh et al., 2023) and understanding benchmarks (Fu et al., 2023; Wang et al., 2025), showcasing its potential for multi-modal reasoning.

## 2 PRELIMINARY: THE UNIFED MODEL - BAGEL

To enable unified CoT across both image and text modalities, our framework builds upon **BAGEL** (Scalable Generative Cognitive Model) (Deng et al., 2025), a unified model that supports joint vision and language generation. We briefly introduce the core elements of BAGEL next.

**Architecture.** BAGEL adopts a decoder-only transformer architecture with a Mixture-of-Transformer-Experts design (Liang et al., 2025). It consists of two experts: an image understanding expert and an image generation expert, which share the same architecture but have different parameters. These experts are bridged by a unified self-attention mechanism over shared multi-modal token sequences. This design enables flexible fusion across modalities without task-specific bottlenecks.

Specifically, BAGEL incorporates two visual encoders for image understanding and generation:

▷ Vision Transformer (ViT): The ViT serves as an image understanding encoder and is initialized from SigLIP2 (Tschannen et al., 2025), ensuring the preservation of rich semantic information. It converts an input image into a $70 \times 70$ latent grid, yielding 4,900 discrete tokens for understanding.

▷ Variational Autoencoder (VAE): The VAE serves as the backbone for image generation and is initialized from FLUX (Labs et al., 2025), enabling the preservation of fine-grained visual details. It maps an input image into a $64 \times 64$ latent grid, producing 4,096 discrete tokens for generation.

With the above designs, BAGEL employs an expert-routing mechanism that dynamically switches between experts and visual encoders for interleaved multi-modal generation:

▷ For image understanding, the understanding expert activates the ViT encoder to convert images into tokens, which are fused with text to enable grounded next-token prediction for tasks such as visual question answering and scene interpretation.

▷ For image generation, the generation expert takes text and image tokens, applies a Rectified Flow process to predict VAE latent features, and decodes them into high-quality images, supporting both image generation and image editing.

This dual-pathway design enables fine-grained visual generation and robust semantic grounding within a unified modeling framework, allowing BAGEL to handle diverse tasks under a single model.

**High Complexity for Multi-Modal Reasoning.** Although BAGEL provides a unified inference, it encounters severe computational bottlenecks when applied to step-wise multi-modal reasoning. Unlike text-only CoT, where each step typically consumes around 300 tokens for text generation, multi-modal CoT requires both image understanding and image generation within each step, resulting in substantial overhead. Specifically, generating an image through the VAE introduces approximately 4,096 tokens, while encoding an image with the ViT for understanding adds another 4,900 tokens. This amounts to nearly 9,000 visual tokens on top of the 300 text tokens per reasoning step, making both training and inference prohibitively expensive. Moreover, such a token-intensive formulation poses significant challenges for optimization, often hindering convergence and limiting generalization, especially in complex tasks that demand long, compositional reasoning chains.

## 3 METHOD

### 3.1 OVERVIEW OF PROBLEM AND OUR SOLUTION

**Problem Setup** Given a multi-modal input $x$, our objective is to generate a multi-modal Chain-of-Thought (CoT) trajectory, an ordered sequence of states $\tau = \{h_0, h_1, \ldots, h_T\}$, to derive a final answer $y$. Each intermediate state $h_t$ can contain both text and images.

Normally, a MLLM $\pi$, parameterized by $\theta$, generates this CoT trajectory autoregressively as:

$$h_t \sim \pi_\theta(\cdot \mid x, h_{<t}), \quad t = 0, \ldots, T, \tag{1}$$

where $h_{<t}$ denotes the history $\{h_0, \ldots, h_{t-1}\}$. Then the final answer is decoded from the terminal state, $y = \text{Dec}(h_T)$. This monolithic generation process requires each step to attend to the entire

history, leading to a quadratic complexity:

$$\mathcal{C}_{\text{raw}}(T) = \sum_{t=1}^{T} \mathcal{O}(t) = \mathcal{O}(T^2). \tag{2}$$

While this is tractable for text-only CoT, where each state $h_t$ consumes fewer than 300 tokens, it becomes prohibitive in the multi-modal setting. Here, a single state includes both textual and visual tokens, resulting in $|h_t| \approx 10{,}000$ (see Section 2), which magnifies the quadratic cost and makes naive autoregressive modeling intractable.

**Our Solution.** To address the quadratic complexity, we introduce a hierarchical framework that decomposes the reasoning trajectory into a high-level **macro-level** process and a fine-grained **micro-level** process. As shown in Figure 2, this framework comprises three main components: a *macro planner*, a set of *micro operators*, and a *macro summarizer*.

▷ *Macro Planner.* Firstly, the planner generates a high-level plan $z_{\text{plan}} = \{z_1\}_1^M$ that decompose the given $x$ to $M$ subgoals. Each subgoal $z_i$ specifies *what* problem to solve, deferring the *how* to the operators. The plan is generated autoregressively as:

$$z_{\text{plan}} = \{z_i\}_{i=1}^{M}, \text{ where } z_i \sim \pi_\theta(\cdot \mid s_0, z_{<i}), \tag{3}$$

where $\pi_\theta$ denotes the sequential macro-policy.

▷ *Micro Operator.* For each subgoal $z_i$, an operator executes a localized reasoning trajectory $\tau_i$ to produce an intermediate result $y_i$. This local trajectory $\tau_i$ is conditioned on the subgoal $z_i$ and optionally the last subgoal's result $y_{i-1}$. It is generated autoregressively as:

$$\tau_i = \{h_i^0, h_i^1, \cdots, h_i^{T_i}\}, \text{ where } h_i^t \sim \pi_\theta(\cdot \mid z_i, y_{i-1}, h_i^{<t}). \tag{4}$$

The subgoal's result is then decoded from the final state of the local trajectory as $y_i = \text{Dec}(s_{i,T_i})$.

▷ *Macro Summarizer.* Finally, the summarizer aggregates the results from all subtasks $\{y_1, \ldots, y_M\}$ to produce the final answer $y$ as :

$$z_{sum} \sim \pi_\theta(\cdot \mid x, z_{\text{plan}}, \{z_i, y_i\}_{i=1}^{M}), \text{ and } y = \text{Dec}(z_{\text{sum}}). \tag{5}$$

In summary, this hierarchical decomposition transforms a single, long-history chain-of-though $\tau = \{h_0, h_1, \ldots, h_T\}$ into a series of modular, localized reasoning blocks:

$$\tau_{\text{hier}} = \{z_{\text{plan}} \oplus (\tau_1 \oplus \cdots \oplus \tau_M) \oplus z_{\text{sum}}\}, \text{ where } \tau_i = \{h_i^0, h_i^1, \cdots, h_i^{T_i}\}. \tag{6}$$

By partitioning the full trajectory of length $T$ into $M$ sub-trajectories, each with average length $\bar{T} \approx T/M$, the quadratic cost is reduced as:

$$\mathcal{C}_{\text{hier}}(T) = \mathcal{O}\big(M \cdot \bar{T}^2\big) = \mathcal{O}\big(M \cdot (T/M)^2\big) = \mathcal{O}(T^2/M), \tag{7}$$

with only minor overhead from high-level planning and summarization. Moreover, by adopting a Markovian transition design at the micro level (Section 3.3), where each state depends only on the previous state and current instruction, the cost is further reduced to $\mathcal{O}(T)$, thereby enabling efficient and scalable multi-modal reasoning. We detail all components in the following sections.

### 3.2 MACRO-LEVEL REASONING: PLANNING AND SUMMARIZATION

Drawing inspiration from human cognition, where complex problems are addressed by first sketching a plan of subgoals and then integrating their outcomes into a final solution (Epstein et al., 2017; Constantinescu et al., 2016), our macro-level CoT adopts a two-phase structure: a planner for task decomposition and a summarizer for evidence integration. We describe how macro-level CoT governs multi-modal reasoning in the following.

**Macro-level Trajectory.** Here, we consider two mechanisms for macro-level reasoning:

▷ *Sequential Decomposition.* Humans often approach difficult problems step by step, solving intermediate goals in sequence. Analogously, our planner generates a macro plan $z_{\text{plan}}$ that decomposes a task $x$ into subgoals $\{z_i\}_{i=1}^{M}$ autoregressively, as described in Equation 3.

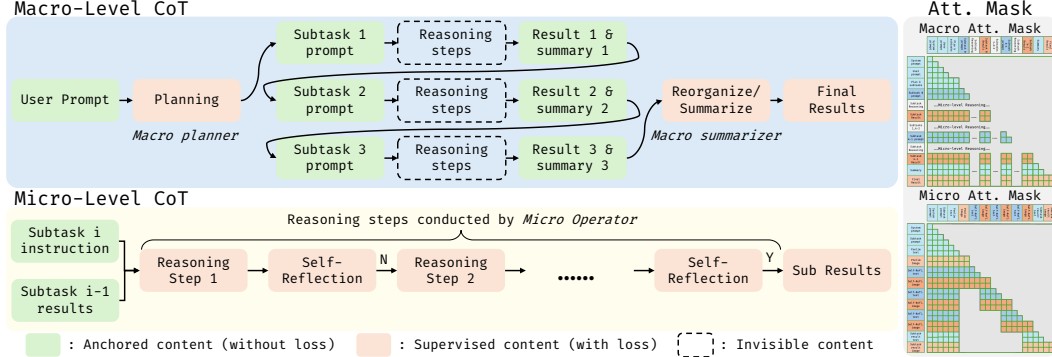

Figure 2: Overview of the Uni-CoT framework. Uni-CoT integrates two reasoning branches: (1) **Macro-Level CoT**, which decomposes a task into subtasks and synthesizes their outputs, enforced by a macro attention mask that reveals only the prompt, plans, and subtask results; (2) **Micro-Level CoT**, which executes each subtask as a Markov Decision Process (MDP), with self-reflection steps conditioned solely on the previous state and current instruction, enforced by a micro attention mask that restricts visibility. High-resolution depictions of both masks are shown in Figure S1.

Under the sequential decomposition mechanism, each subgoal is executed only after the previous one has been completed. The resulting hierarchical reasoning trajectory is formulated as

$$\tau_{\text{hier}} = \big\{ z_{\text{plan}} \oplus (\tau_1 \oplus \cdots \oplus \tau_M) \oplus z_{\text{sum}} \big\}, \tag{8}$$

where $\oplus$ denote concatenate. Then the micro-CoT $\tau_i = \{h_i^t\}_{t=0}^{T_i}$ is initialized from the subgoal $z_i$ and last subgoal's result $y_{i-1}$, and is generated sequentially as

$$\begin{cases} h_t^i = \pi_\theta(z_i, y_{i-1}), & t = 0 \\ h_t^i = \pi_\theta(z_i, h_i^{<t}), & t \geq 1. \end{cases} \tag{9}$$

Finally, the macro summarization state is defined as the last subtask output as $z_{\text{sum}} = y_M$, and the final answer is obtained by decoding as $y = \text{Dec}(z_{\text{sum}})$.

▷ *Parallel Decomposition.* Alternatively, humans may accelerate problem solving by dividing a task into independent components that can be solved in parallel. Inspired by this, we introduce a parallel decomposition strategy where subgoals are executed concurrently. The plan generation follows the same procedure as above, but the hierarchical trajectory is now formulated as

$$\tau_{\text{hier}} = \big\{ z_{\text{plan}} \oplus (\|_{i=1}^M \tau_i) \oplus z_{\text{sum}} \big\}, \tag{10}$$

where $\|_{i=1}^M$ denotes parallel execution. The $\tau_i$ initialize its state only on its assigned subgoal as $h_0^t = \pi_\theta(z_i)$ and follow Equation 9 to conduct whole micro-level CoT. Then the macro summarizer integrates all outcomes to synthesis final answer as:

$$z_{\text{sum}} \sim \pi_\theta(\cdot \mid x, \{z_i, y_i\}_{i=1}^M), \qquad y = \text{Dec}(z_{\text{sum}}). \tag{11}$$

**Macro Attention Mask.** In macro-level reasoning, the focus lies on planning the overall task and summarizing the outcomes of all subtasks to synthesize the final result, while abstracting away low-level operations (i.e., the micro-level traces of completed subtasks) to reduce computational burden. To enforce this abstraction, we introduce a *macro attention mask* (Figure 2 or Figure S1(a)), which restricts visibility during planning and summarization to macro-level states only: the input, subgoals, intermediate results, and the final answer.

### 3.3 MICRO-LEVEL REASONING: MDP-GUIDED SELF-REFLECTION

Once a subgoal is assigned by the macro planner, the micro-level CoT handles its execution. As the overall reliability of the system depends on the coherence of outputs across subtasks, the core objective of the micro-level CoT is to produce accurate and reliable results for the assigned subgoal.

To achieve this, we introduce a *Self-Reflection* mechanism that improves robustness and adaptability during subtask execution. As illustrated in Figure 2, after completing an initial attempt, the model

evaluates the quality of its output and determines whether revision is necessary. If logical inconsistencies or cross-modal mismatches are detected, the model revises the output and re-evaluates it in a closed-loop feedback cycle. We detail this mechanism below.

**Micro-Level Trajectory.**  The reasoning trajectory for subgoal $z_i$ is represented as

$$\tau_i = \{h_i^0 \to h_i^1 \to \cdots \to h_i^{T_i}\}, \text{ where } h_i^t \sim \pi_\theta(\cdot \mid z_i, , h_i^{<t}) \text{ for } t > 0. \tag{12}$$

The trajectory is initialized as $h_0^i \sim \pi_\theta(z_i, y_{i-1})$ or $\pi_\theta(z_i)$, reflecting the model's initial attempt to solve the subtask. Subsequent states refer to the revision steps through self-reflection.

At each self-reflection step, the model performs three operations: (i) generate an evaluation score $eval_t^i$ for the current state, (ii) decide whether further refinement is needed, and (iii) apply refinement if necessary, including textual editing prompt generation $rev\_txt_i^{t+1}$ and corresponding image editing $rev\_img_i^{t+1}$. This can be formalized as

$$\tau_i = \{(h_i^t \to eval_i^t \to rev\_txt_i^{t+1} \to rev\_img_i^{t+1} \to h_{t+1}^i)\}_{t=0}^{T_i-1}. \tag{13}$$

The process continues until the evaluation score exceeds a threshold, yielding the final trajectory $\tau_i$.

As described above, self-reflection process can be naturally formulated as a Markov Decision Process (MDP) $\{(s_t^i, a_t^i, r_{t+1}^i, s_{t+1}^i)\}_{t=0}^{T_i-1}$: (1) state $s_t^i$ is the reasoning state $h_i^t$; (2) action $a_t^i$ represents the refinement action $[rev\_txt_i^{t+1}, rev\_img_i^{t+1}]$; and (3) reward $r_{t+1}^i$ is the evaluation score $eval_t^i$.

A key difference remains between the raw trajectory and the MDP formulation: in the raw trajectory, each new state attends to the entire history, whereas in the MDP formulation, the transition $s_t^i \to s_{t+1}^i$ depends only on the current state and the fixed subgoal $z_i$. Analogous to human self-reflection, where one typically focuses on the present state to revise work rather than recalling the full history, we further enforce this constraint by masking the attention layer. Thus, the current state is generated only from the last state and the subgoal: $h_t^i \sim \pi_\theta(\cdot \mid z_i, h_{t-1}^i)$.

**Complexity Discussion.**  If self-reflection is directly modeled on the raw CoT trajectory, each new state $h_i^t$ attends to the full history $h_i^{<t}$, leading to a quadratic complexity $\mathcal{O}(T_i^2)$. By instead casting the process as a MDP, the transition depends only on the last state, yielding linear cost:

$$\mathcal{C}_{\text{mdp}}(T_i) = \mathcal{O}(T_i). \tag{14}$$

Recalling Equation 3, the macro–micro structure decomposes a trajectory of length $T$ into $M$ sub-trajectories of length $\bar{T} \approx T/M$, reducing the complexity from $\mathcal{O}(T^2)$ to $\mathcal{O}(T^2/M)$. With the MDP formulation applied to each sub-trajectory, the overall cost further reduces near-linear complexity:

$$\mathcal{C}_{\text{hier-mdp}}(T) = \mathcal{O}(M \cdot T/M) = \mathcal{O}(T). \tag{15}$$

This efficiency gain enables scaling self-reflective reasoning to long multi-modal trajectories.

**Micro Attention Mask.**  To enforce locality, we introduce a *micro attention mask* that restricts visibility to $(s_{t-1}^i \to s_t^i)$ and the fixed subgoal $z_i$, while masking unrelated history as shown in Figure 2 or Figure S1(b). This design simplifies optimization and improves computational efficiency.

## 3.4 TRAINING PARADIGM

The Uni-CoT training contains two parts: Macro-Level CoT learning and Micro-Level CoT learning. We briefly introduce both in below and more details are provided in Appendix Section C.

For the Macro-Level CoT, we follow (Deng et al., 2025) to use a joint loss that supervises both text and image generation for planning and final synthesis. Specifically, we apply cross-entropy (CE) loss for text generation and mean squared error (MSE) loss for image generation:

$$\mathcal{L}_{\text{joint}} = \lambda_{\text{CE}} \cdot \mathcal{L}_{\text{CE}}^{\text{text}} + \mathcal{L}_{\text{MSE}}^{\text{image}}, \tag{16}$$

where $\lambda$ is a balancing coefficient that controls the relative importance of textual and visual losses.

For the Micro-Level CoT, subtask completion is supervised using interleaved multi-modal data, similarly adopting the joint loss $\mathcal{L}_{\text{joint}}$ for model learning. Additionally, to learn MDP-based self-reflective process, we decompose learning into four auxiliary objectives: text action $a_t^{text}$ generation, image action generation, next-state prediction and reward estimation.

## 4 EXPERIMENTS

### 4.1 IMPLEMENTATION DETAILS

**Dataset.** We construct a multi-modal reasoning dataset containing approximately 30K samples to support both macro- and micro-level Chain-of-Thought (CoT) learning in Uni-CoT. Starting from curated text or text–image prompts, we augment each prompt with explicit logical deductions and enriched visual details. The enhanced prompts are then decomposed into 2-3 subtasks using large language models (LLMs), forming macro-level planning traces. Image understanding and generation models then are utilized to complete each subtask and iterative self-reflection, producing fine-grained micro-level trajectories. The final dataset includes 11K interleaved text–image pairs for macro-level planning and 20K for micro-level refinement. We also crop data from GeoPose3K (Brejcha & Čadík, 2017), PIPE (Wasserman et al., 2025), Sharegpt-4o-image (Chen et al., 2025b), Echo-4o (Ye et al., 2025) to enhance the basic image understanding and generation of our model. For further detail please refer to Appendix Section D.

**Training Setup.** We build Uni-CoT on top of the unified model Bagel (Deng et al., 2025) and fine-tune it on our 31k-sample CoT dataset. Training is supervised using a combination of cross-entropy loss for textual reasoning and mean-squared error loss for visual reasoning. With the hierarchical design, Uni-CoT is trained efficiently on 8× NVIDIA A100 GPUs within one week. We optimize all parameters with Adam using a constant learning rate of 2e-5 and apply a linear warmup during the first 200 steps. At each iteration, we mix all data sources, understanding, generation, and transition traces, uniformly sample examples, and pack them into sequences of up to 32,768 tokens. Additional training details are provided in Appendix Section C.

### 4.2 EXPERIMENTAL SETUP

In the main paper, we focus on evaluating Uni-CoT on two core tasks: image generation and image understanding. **Ablation study** and **more experiments** are deferred to Appendix Section E.

For the image generation task, following Deng et al. (2025), we conduct experiments on two widely adopted benchmarks: GenEval (Ghosh et al., 2023) and WISE (Niu et al., 2025). GenEval (Ghosh et al., 2023) serves as a general benchmark for assessing object-focused text-to-image alignment, while WISE (Niu et al., 2025) is a reasoning-driven benchmark designed to evaluate a model's ability to generate faithful outputs from abstract, reasoning-intensive prompts.

For image understanding, we evaluate Uni-CoT on general multi-modal reasoning benchmarks, including MME (Fu et al., 2023), MMMU (Yue et al., 2024), MathVista (Lu et al., 2023), MM-Bench (Liu et al., 2024b). We further evaluate Uni-CoT on Jigsaw-R1 (Wang et al., 2025), a structured multi-modal reasoning benchmark involving jigsaw-puzzle solving across varying difficulty levels, testing a model's ability on structured visual reasoning.

### 4.3 RESULTS FOR IMAGE GENERATION

**Quantitative Results.** Table 1 and Table 2 shows results on GenEval (Ghosh et al., 2023) and WISE (Niu et al., 2025), benchmarking basic and reasoning-based image generation, respectively. Uni-CoT surpasses its base model Bagel on GenEval, with improvements largely, mainly relying on our macro decomposition strategy. On WISE, Uni-CoT achieves state-of-the-art performance across all domains, demonstrating substantially stronger reasoning-driven generation compared to open-source baselines due to self-reflection mechanism that corrects initial errors.

**Qualitative Analysis.** As shown in the first row of Figure 3, Uni-CoT generates a coherent planning strategy that transforms a challenging, unnatural GenEval prompt into a sequence of natural intermediate prompts, enabling smoother image generation. In contrast, the second row of Figure 3 illustrates the model's ability to correct semantically inaccurate outputs through multi-round self-reflection. By integrating these two mechanisms, Uni-CoT achieves reliable and significant improvements across benchmarks. Additional visualization results are provided in Appendix E.

Table 1: Quantitative evaluation results on GenEval (Ghosh et al., 2023). Uni-CoT w/o CoT is our initial results without multi-modal reasoning.

| Type | Model | Single Obj. | Two Obj. | Counting | Colors | Position | Color Attri. | Overall↑ |
|---|---|---|---|---|---|---|---|---|
| *Gen. Only* | PixArt-$\alpha$ Chen et al. (2024a) | 0.98 | 0.50 | 0.44 | 0.80 | 0.08 | 0.07 | 0.48 |
| | DALL-E 2 Ramesh et al. (2022) | 0.94 | 0.66 | 0.49 | 0.77 | 0.10 | 0.19 | 0.52 |
| | Emu3-Gen Wang et al. (2024b) | 0.98 | 0.71 | 0.34 | 0.81 | 0.17 | 0.21 | 0.54 |
| | SDXL Podell et al. (2024) | 0.98 | 0.74 | 0.39 | 0.85 | 0.15 | 0.23 | 0.55 |
| | DALL-E 3 Betker et al. (2023) | 0.96 | 0.87 | 0.47 | 0.83 | 0.43 | 0.45 | 0.67 |
| | SD3-Medium Esser et al. (2024) | 0.99 | 0.94 | 0.72 | 0.89 | 0.33 | 0.60 | 0.74 |
| | FLUX.1-dev Labs (2024) | 0.98 | 0.93 | 0.75 | 0.93 | 0.68 | 0.65 | *0.82* |
| *Unified* | LWM Liu et al. (2024a) | 0.93 | 0.41 | 0.46 | 0.79 | 0.09 | 0.15 | 0.47 |
| | SEED-X Ge et al. (2024) | 0.97 | 0.58 | 0.26 | 0.80 | 0.19 | 0.14 | 0.49 |
| | TokenFlow-XL Qu et al. (2024) | 0.95 | 0.60 | 0.41 | 0.81 | 0.16 | 0.24 | 0.55 |
| | ILLUME Wang et al. (2024a) | 0.99 | 0.86 | 0.45 | 0.71 | 0.39 | 0.28 | 0.61 |
| | Emu3-Gen Wang et al. (2024b) | 0.99 | 0.81 | 0.42 | 0.80 | 0.49 | 0.45 | 0.66 |
| | Show-o Xie et al. (2024) | 0.98 | 0.80 | 0.66 | 0.84 | 0.31 | 0.50 | 0.68 |
| | Janus-Pro-7B Chen et al. (2025c) | 0.99 | 0.89 | 0.59 | 0.90 | 0.79 | 0.66 | 0.80 |
| | MetaQuery-XL Pan et al. (2025) | - | - | - | - | - | - | 0.80 |
| | BAGEL Deng et al. (2025)[†] | 0.99 | 0.92 | 0.78 | 0.87 | 0.53 | 0.64 | 0.79 |
| | **Uni-CoT w/o CoT** | 0.99 | 0.95 | 0.82 | 0.90 | 0.55 | 0.69 | 0.81 |
| | **Uni-CoT** | 0.99 | 0.96 | 0.84 | 0.92 | 0.57 | 0.71 | **0.83** |

Table 2: Quantitative evaluation results on WISE (Niu et al., 2025). Uni-CoT w/o CoT is our initial results without multi-modal reasoning.

| Model | Culture | Time | Space | Biology | Physics | Chemistry | Overall↑ |
|---|---|---|---|---|---|---|---|
| Janus Wu et al. (2024a) | 0.16 | 0.26 | 0.35 | 0.28 | 0.30 | 0.14 | 0.23 |
| MetaQuery Pan et al. (2025) | 0.56 | 0.55 | 0.62 | 0.49 | 0.63 | 0.41 | 0.55 |
| Bagel-Think Deng et al. (2025) | **0.76** | 0.69 | 0.75 | 0.65 | 0.75 | 0.58 | 0.70 |
| *GPT-4o Hurst et al. (2024)* | *0.81* | *0.71* | *0.89* | *0.83* | *0.79* | *0.74* | *0.80* |
| **Uni-CoT w/o CoT** | 0.75 | 0.67 | 0.79 | 0.64 | 0.79 | 0.65 | 0.72 |
| **Uni-CoT** | **0.76** | **0.70** | **0.76** | **0.73** | **0.81** | **0.73** | **0.75** |

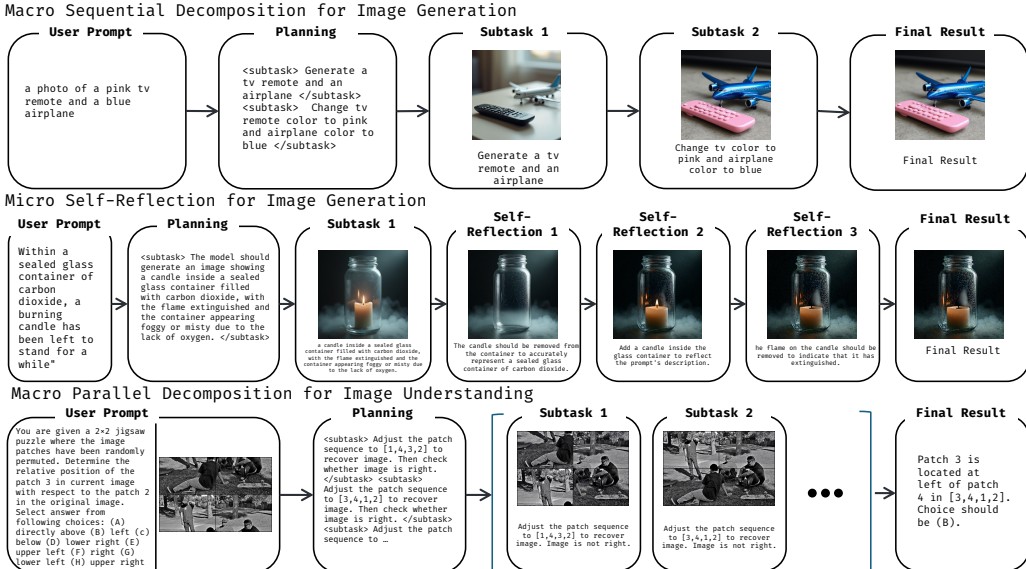

Figure 3: Visualization of the Uni-CoT multi-modal reasoning process: macro-level decomposition (top), micro-level self-reflection (middle), macro-level parallel decomposition (bottom).

## 4.4 RESULTS FOR IMAGE UNDERSTANDING

**Quantitative Results.** We report results on the general multi-modal understanding benchmarks and Jigsaw-R1 benchmarks in Table 3. On the general benchmarks (including MME, MathVista and MMBench), Uni-CoT achieves slightly higher scores than its base model Bagel, indicating that it preserves broad world knowledge while also benefiting from our post-training strategy. In contrast, on the Jigsaw-R1 benchmark, Uni-CoT substantially outperforms all compared open-sourced mod-

Table 3: Image understanding results on MME, MMMU, MMBench, MathVista, and Jigsaw-R1.

| Method | General Benchmarks | | | | | Jigsaw-R1 | | | | |
|---|---|---|---|---|---|---|---|---|---|---|
| | MME-P↑ | MME-S↑ | MMMU↑ | MMBench↑ | MathVista↑ | 2×1 | 3×1 | 4×1 | 2×2 | Overall↑ |
| *Random* | - | - | - | - | - | 50.00 | 50.00 | 50.00 | 12.50 | 40.63 |
| GPT-4V (Achiam et al., 2023) | *1409* | *1927* | *56.8* | *74.3* | *47.5* | - | - | - | - | - |
| GPT-4.1-mini (OpenAI, 2025) | - | - | - | - | - | *61.90* | *54.50* | *54.80* | *20.30* | *47.88* |
| InternVL2.5-2B (Chen et al., 2024b) | - | 2138 | 38.2 | - | 51.3 | 44.90 | 41.90 | 48.60 | 9.70 | 36.28 |
| Qwen2.5-VL-7B (Bai et al., 2025) | - | 2347 | **58.6** | 83.5 | 68.2 | 49.40 | 48.70 | 50.60 | 15.80 | 41.12 |
| BAGEL (Deng et al., 2025) | **1687** | 2388 | 52.8 | 85.0 | 73.1 | **51.25** | 50.05 | 52.05 | 9.60 | 40.73 |
| **Uni-CoT** | **1687** | 2392 | 52.7 | **85.6** | **73.3** | 51.15 | **61.46** | **59.15** | **18.64** | **47.60** |

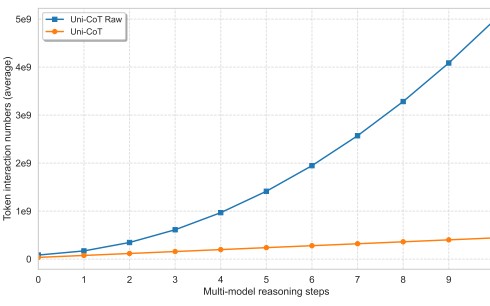

Figure 4: Complexity comparison.

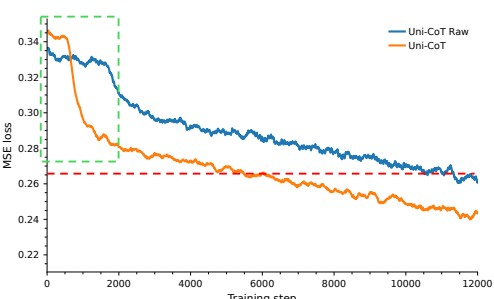

Figure 5: MSE loss comparison.

els, demonstrating its strong multi-modal reasoning capabilities, particularly in perception-heavy tasks.

**Qualitative Analysis.** The bottom row of Figure 3 showcases the parallel decomposition mechanism for solving a Jigsaw-R1 case. By examining candidate options concurrently, Uni-CoT can efficiently process distinct puzzle segments in parallel and subsequently integrate their results into a coherent solution. This strategy accelerates convergence and reduces error propagation across subtasks, underscoring the advantages of parallel decomposition in perception-heavy reasoning tasks.

## 4.5 COMPLEXITY ANALYSIS

In Section 3.3, we theoretically analyze that our hierarchical design can reduce the complexity from quadratic to near-linear. Here, we empirically validate the effectiveness of our method by measuring the actual inference cost of two variants on 100 samples: (1) a naïve Uni-CoT without the proposed hierarchical framework (Uni-CoT Raw), (2) the complete Uni-CoT (Uni-CoT).

As shown in Figure 4, Uni-CoT Raw consumes substantially more tokens than Uni-CoT and incurs an almost quadratic increase in computation as the number of reasoning steps grows. In contrast, Uni-CoT requires far fewer tokens and exhibits near-linear scaling. This quadratic-versus-near-linear gap confirms the significant complexity reduction achieved by our hierarchical, interleaved reasoning framework. Notably, with 2 and 3 multi-modal reasoning steps, Uni-CoT reduces the average token interaction count by 2.24× and 2.95×, respectively, and the reduction grows to 11.26× when the number of reasoning steps reaches 10.

We further examine the training dynamics of Uni-CoT in comparison with Uni-CoT Raw. As illustrated in Figure 5, Uni-CoT converges substantially faster, reaching a comparable loss level within only 6,000 training steps, whereas Uni-CoT Raw requires 12,000 steps. When both models are trained for 12,000 steps on a small dataset of approximately 10,000 samples, Uni-CoT consistently outperforms Uni-CoT Raw on WISE (0.73 vs. 0.70), indicating not only improved training efficiency but also superior output quality. More details refer to Appendix Section E.

## 5 RELATED WORK

**Multi-modal CoT with Visual Reasoning.** Recent work has extended CoT to MLLMs to better align reasoning with visual content (Zhang et al.; Mitra et al., 2024; Zheng et al., 2023). RL-based approaches improve textual reasoning (Zhang et al., 2025a; Yang et al., 2025b; Feng et al., 2025) but struggle in visually intensive domains (Wu et al., 2024b; Cheng et al., 2025). To approximate

visual state transitions, some methods introduce programmatic operations—cropping, drawing, or plotting (Gupta & Kembhavi, 2023; Surís et al., 2023; Hu et al., 2024a;b; Shen et al., 2025; Gao et al., 2025), which capture local changes but fail to model the global structural transformations required for navigation or rearrangement tasks. Another direction couples MLLMs with image or video generators to support larger-scale visual transitions for image understanding (Zhao et al., 2025b) or image generation (Guo et al., 2025a; Xue et al., 2025; Zhang et al., 2025c). Others refine outputs using external reward models (Guo et al., 2025b; Jiang et al., 2025). However, these loosely coupled designs often yield fragmented reasoning flows, limiting overall coherence.

**Hierarchical reasoning,** where high-level planning guides low-level execution, is well established in cognitive science (Epstein et al., 2017; Orru & Longo, 2018). Inspired by this principle, prior work has explored hierarchical structures in text-only CoT (Yang et al., 2025a; Wu et al., 2025a), hierarchical RL (Vezhnevets et al., 2017; Nachum et al., 2018), and multi-agent planning (Zhang et al., 2025b). However, these efforts primarily aim to improve reasoning quality, controllability, or planning efficiency. In contrast, our work leverages hierarchy for a fundamentally different purpose: to reduce the training and inference complexity inherent to multi-modal CoT reasoning while preserving the model's reasoning ability. This focus on complexity reduction in multi-modal settings distinguishes our approach from prior hierarchical methods.

**Efficient Attention** is an important and widely studied research direction (Sun et al., 2025), aiming to reduce the computational cost of transformers. Prior methods primarily focus on lowering attention complexity through sparsification (Child et al., 2019), low-rank approximations (Wang et al., 2020; Tay et al., 2020), or locality-biased attention (Liu et al., 2021), nearly without consideration for the model's reasoning ability. In contrast, our formulation is driven by the need to reduce the reasoning complexity itself while preserving the integrity of CoT reasoning, making our approach inherent different from prior efficient-attention designs.

## 6 CONCLUSION

We present Uni-CoT, a unified Chain-of-Thought framework that enables coherent and grounded multimodal reasoning across vision and language within a single model. By introducing a two-level hierarchical reasoning architecture, we significantly reduce computational complexity and improve reasoning efficiency. Extensive experiments across reasoning-driven image generation and editing benchmarks demonstrate Uni-CoT's superiority in both performance and interpretability. We believe Uni-CoT offers a scalable foundation for future multi-modal reasoning systems.

While Uni-CoT achieves strong results on generation and understanding tasks, extending the framework to broader real-world applications, particularly those requiring fine-grained visual consistency or perfectly text–image alignment, remains an open challenge. We regard this as a promising direction for future work, where improved trajectory modeling and stronger visual–textual alignment could further expand the capability and applicability of Uni-CoT.

## 7 ACKNOWLEDGMENT

This work was supported by AI for Science Program, Shanghai Municipal Commission of Economy and Informatization (Grant No. 2025-GZL-RGZN-BTBX-02017).

We thank Zhengbo Zhang, Li Xu, Qi Lv, Jun Gao, Qian Qiao and Zhiheng Li for their valuable discussions and constructive feedback.

## 8 REPRODUCIBILITY STATEMENT

We are committed to ensuring the reproducibility of our work. To this end, we provide detailed descriptions of the model architecture, training objectives, datasets, preprocessing steps, and evaluation protocols in the appendix. Most of hyperparameter settings and implementation details are documented. Moreover, we have released the codebase together with scripts for inference, to facilitate independent verification of our framework and future research.

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

APPENDIX CONTENT

## A   THE USE OF LARGE LANGUAGE MODELS (LLMS)

In preparing this paper, we made limited use of a Large Language Model (LLM) as a general-purpose writing assistant. Specifically, the LLM was employed to polish the language, improve clarity, and optimize the organization of certain sections. It did not contribute to the core research process, including problem formulation, experimental design, data analysis, or interpretation of results.

## B   DETAILS OF ATTENTION MASK

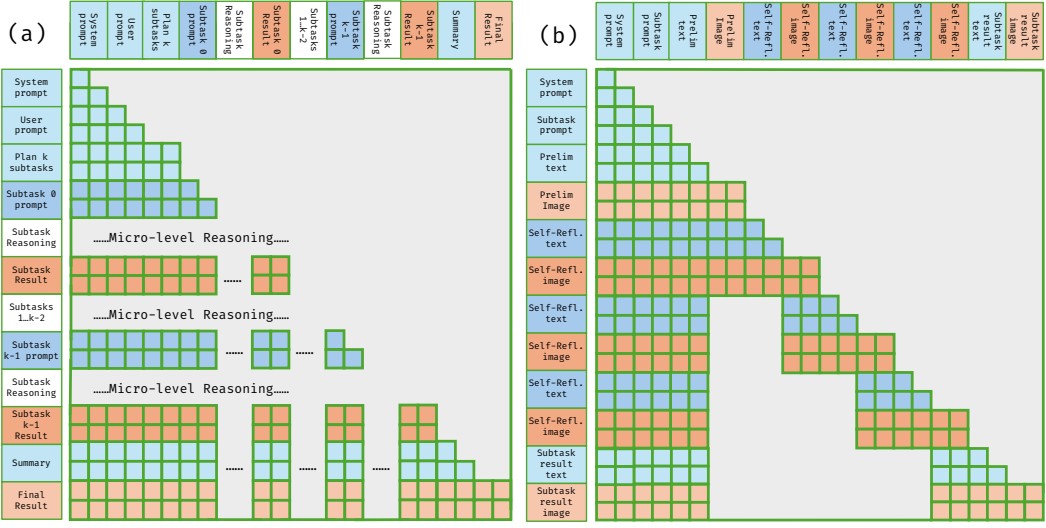

Figure S1: Attention mask: (a) Macro attention mask; (b) Micro attention mask.

As shown in Figure S1(a), under the Macro attention mask scheme, the model has access only to the system prompt, high-level planning outputs, and final subtask outcomes, while intermediate reasoning traces, such as textual rationales or image edits generated during subtask execution, are

completely masked. In contrast, Figure S1(b) illustrates the Micro attention mask scheme, where each self-reflection step can attend only to the immediately preceding state (e.g., the last image-text pair) and the current subtask instruction.

# C   DETAILS OF TRAINING

## C.1   TRAINING PARADIGM

We decompose Uni-CoT training into two reasoning levels: **Macro-Level CoT learning**, which captures global planning and final result synthesis, and **Micro-Level CoT learning**, which focuses on subtask execution and self-reflection.

**Macro-Level CoT Learning.**   The macro-level reasoning process consists of two components: *a planner for task decomposition* and *a summarizer for evidence integration..* For the planner, the training task is framed as an understanding problem, where the model takes in multi-modal inputs and then produces a structured textual plan, trained under a cross-entropy loss $\mathcal{L}_{\mathrm{CE}}^{\mathrm{txt}}$. For the summarizer, the training task is cast as a generative problem: the model integrates both the global plan and the intermediate multi-modal outputs to generate the final solution. For understanding tasks, this corresponds to producing a textual conclusion (supervised by cross-entropy loss $\mathcal{L}_{\mathrm{CE}}^{\mathrm{txt}}$), whereas for image generation tasks, it corresponds to synthesizing both textual and visual outputs (supervised joint loss $\mathcal{L}_{\mathrm{joint}}$ that combines cross-entropy (CE) and mean squared error (MSE) losses). The data structures underlying these two stages are summarized in Table S1.

Specifically, for image understanding, we define the CE loss $\mathcal{L}_{\mathrm{CE}}^{\mathrm{txt}}$ as:

$$\mathcal{L}_{\mathrm{CE}}^{\mathrm{txt}} = -\sum_{i=1}^{C} x_i \log(\hat{x}_i), \tag{S1}$$

where $x_i$ denotes the ground-truth token, $\hat{x}_i$ represents the predicted probability of that token, and $C$ is the vocabulary size.

For image generation, we first review the Rectified Flow (Liu et al.) paradigm and then give the MSE loss definition. For denoising-based generation under the Rectified Flow (Liu et al.) paradigm, given a clean latent $\mathbf{x}_0$ and a Gaussian noise sample $\mathbf{x}_1$, we construct the noisy latent $\mathbf{x}_t$ via linear interpolation:

$$\mathbf{x}_t = (1-t) \cdot \mathbf{x}_0 + t \cdot \mathbf{x}_1, \quad t \in [0,1] \tag{S2}$$

The model is then trained to predict the velocity field using the MSE loss:

$$\mathcal{L}_{\mathrm{MSE}}^{\mathrm{image}} = \mathbb{E}\left[\|\pi_\theta(\mathbf{x}_t \mid \mathbf{c}) - (\mathbf{x}_0 - \mathbf{x}_1)\|^2\right] \tag{S3}$$

Here, $\pi_\theta(\mathbf{x}_t \mid \mathbf{c})$ denotes the velocity predicted by the model $\pi$, conditioned on noisy latent $\mathbf{x}_t$ and context $\mathbf{c}$.

Then the joint loss $\mathcal{L}_{\mathrm{joint}}$ is defined as:

$$\mathcal{L}_{\mathrm{joint}} = \lambda \cdot \mathcal{L}_{\mathrm{CE}}^{\mathrm{txt}} + \mathcal{L}_{\mathrm{MSE}}^{\mathrm{img}}, \tag{S4}$$

where $\lambda$ is a coefficient to balance two losses.

**The Micro-Level CoT learning.**   The micro-level reasoning process consists of two complementary components: *subtask completion* and *self-reflection modeling*.

For **subtask completion**, the model is supervised with interleaved multi-modal signals. Specifically, analogous to the final result synthesis, we employ a joint loss $\mathcal{L}_{\mathrm{joint}}$ that integrates cross-entropy loss for textual outputs and mean-squared error loss for image predictions, thereby ensuring alignment across modalities.

For **self-reflection modeling**, as detailed in Sec. 3.3, we formulate the self-reflective procedure within a Markov Decision Process (MDP) framework. Based on this formulation, we introduce four auxiliary objectives for self-reflection modeling:

Table S1: Macro-Level CoT training tasks. Content enclosed in ” ” indicates the target to be learned.

| Objective | Data Structure | Loss |
|-----------|----------------|------|
| **Global Planning** | [System Prompt, Multi-Modal Inputs, "Planning Prompt"] | Cross-Entropy Loss ($\mathcal{L}_{\text{CE}}^{\text{txt}}$) |
| **Result Synthesis** | [System Prompt, Multi-Modal Inputs, Generated Plan, Multi-Modal Intermediate Results, "Final Results"] | Joint Loss ($\mathcal{L}_{\text{joint}}$) |

Table S2: Micro-Level CoT training tasks. Content enclosed in ” ” indicates the target to be learned.

| Objective | Data Structure | Loss |
|-----------|----------------|------|
| **Subtask Completion** | [System Prompt, Subtask Prompt, "Initial Image & Text Results"] | Joint Loss ($\mathcal{L}_{\text{joint}}$) |
| **Text Action** $a_t^{\text{txt}}$ | [System Prompt, Subtask Prompt, Current Image & Text, "Editing Prompt"] | Cross-Entropy Loss ($\mathcal{L}_{\text{CE}}^{\text{txt}}$) |
| **Image Action** $a_t^{\text{img}}$ | [System Prompt, Subtask Prompt, Current Image & Text, Editing Prompt, "Edited Image"] | Mean Square Error Loss ($\mathcal{L}_{\text{MSE}}^{img}$) |
| **Next-State** $s_{t+1}$ | [System Prompt, Subtask Prompt, Edited Image, "Image Analysis"] | Cross-Entropy Loss ($\mathcal{L}_{\text{CE}}^{\text{txt}}$) |
| **Reward** $r_t$ | [System Prompt, Subtask Prompt, Edited Image, Image Analysis, "Evaluation"] | Cross-Entropy Loss ($\mathcal{L}_{\text{CE}}^{\text{txt}}$) |

- *Text Action $a_t^{txt}$ Generation*, where the model evaluates the intermediate results and predicts an editing instruction $a_t^{\text{txt}}$, supervised by cross-entropy loss $\mathcal{L}_{\text{CE}}^{\text{txt}}$;

- *Image Action $a_t^{img}$ Generation*, where the model generates visual modifications $a_t^{\text{img}}$ conditioned on the textual instruction $a_t^{\text{txt}}$, supervised by mean squared error loss $\mathcal{L}_{\text{MSE}}^{\text{img}}$;

- *Next-State $s_{t+1}$ Prediction*, where the model analyzes and summarizes the status of the modified image, supervised by cross-entropy loss $\mathcal{L}_{\text{CE}}^{\text{txt}}$;

- *Reward $r_t$ Estimation*, where the model regresses to a scalar feedback signal or textual description that measures the quality of the intermediate reasoning step relative to the final task objective, supervised by cross-entropy loss $\mathcal{L}_{\text{CE}}^{\text{txt}}$.

The data structure is detailed in Table S2.

## C.2 TRAINING HYPER PARAMETERS

Owing to our decomposition of interleaved image-text CoT via MDP modeling and high-quality data construction, the training of Uni-CoT is quite efficient, as all experiments can be accomplished on 8 NVIDIA A100 GPUs. Following common practices, we utilize FlashAttention, Fully Sharded Data Parallel (FSDP), and mixed-precision training for better computation efficiency. Throughout the training process, all the parameters in the unified model are optimized using Adam optimizer with a constant learning rate of 2e-5. Additionally, we employ a linear warmup learning rate schedule, increasing the learning rate from zero over the first 200 steps. During each training step, we combine all data used for understanding and generation expert training, randomly sample from the combined dataset, and pack the samples into sequences with a target length of 32,768 tokens.

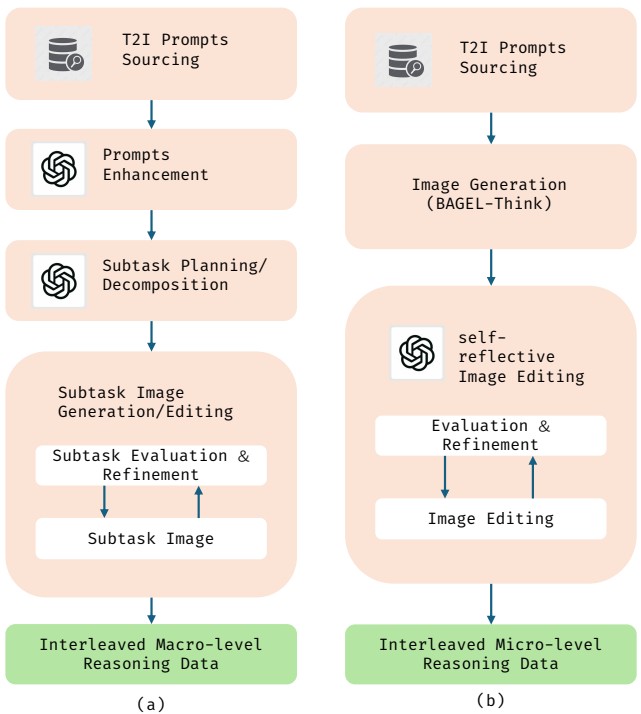

Figure S2: Data curation pipeline for hierarchical reasoning. We collect T2I prompts from various datasets, then for (a)Macro-level reasoning data pipeline: prompts are first enhanced via GPT-4o or Qwen3, then decomposed into several sequential or parallel subtasks. Then GPT-4o as well as Bagel-Think are used for subtask image generation, evaluation and refinement.(b)Micro-level reasoning data pipeline: we use Bagel-Think model to generate preliminary images based on collected T2I prompts. We then perform several rounds of self-reflection, using GPT-4o to first evaluate on current images and generate refinement instruction, then generate edited images conditioned on refinement instruction.

# D  DETAILS OF DATASET.

**Data curation process.**  Figure S2 illustrates the data pipeline we used for data curation. We collect interleaved text-image data for Macro-level and Micro-level reasoning paradigm, respectively. We prepare text-to-image generation prompts from multiple datasets as seed prompts for prompt expansion.

For Macro-level reasoning data, we then enhance the prompts in the following two aspects: (1) We first check if the given prompts entail domain expertise knowledge or common-sense reasoning. If such reasoning or logical induction exist in prompts, we rewrite the prompts to explicitly state the result of reasoning deduction. For example, if the original prompt describes "a melting ice cream cone in desert sun", then the deduced rewritten prompt should elaborate on the object states under given condition, i.e. "dripping, puddle on hot sand" (2) We then enrich the generation prompts via adding auxiliary visual detail or attributes to concretely illustrate any abstract concepts or description, especially those regarding art styles, vibes, environment, etc. After prompt enhancement, we employ models such as GPT-4o or Qwen-plus to decompose the enhanced prompts into 2-3 subtasks such that complex image generation goal inferred by the enhanced prompts are broken into simpler and logically coherent sequential or parallel sub-goal. We then utilize models capable of image generation such as BAGEL-Think or GPT-4o, to perform image generation and editing following the subtask instruction. We also employ VLM models (GPT-4o) to evaluate on the results of subtask excecution and generate corresponding subtask refinement instruction. The textual detail of subtask planning, subtask evaluation and refinement, and the intermediate images generated in all subtask steps, are all collected as interleaved Macro-level data.

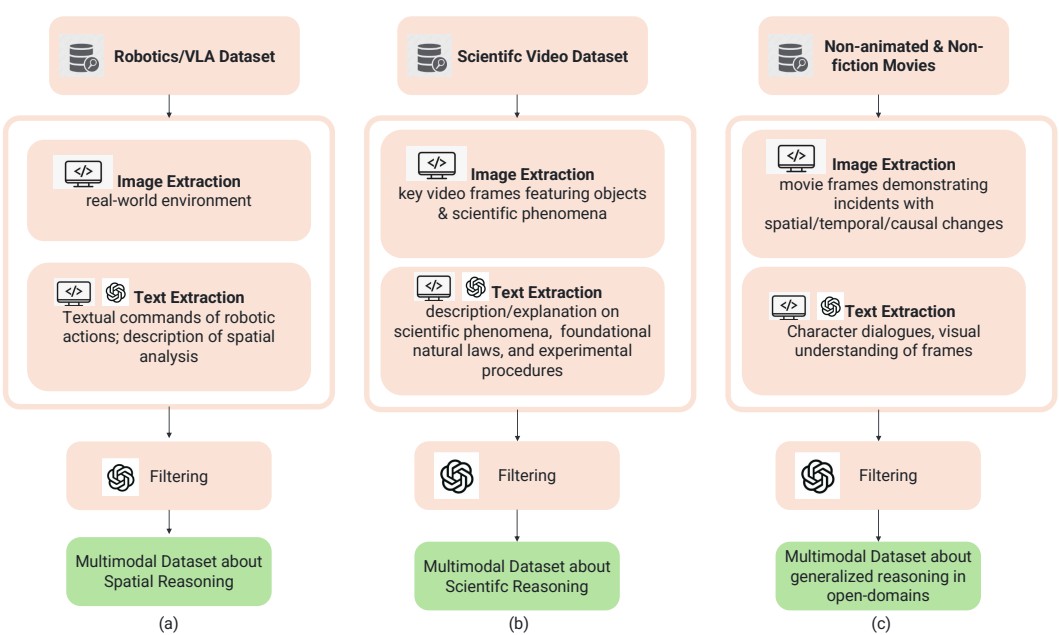

Figure S3: We leverage existing real-world large-scale datasets that embody rich image and textual information and prepare three structured and scalable data curation pipelines across various domains. (a) Images and textual data are extracted from open-source Robotics/VLA datasets to form multi-modal samples that naturally embeds spatial causal relationship and physical interaction reasoning. (b) For scientific and procedural reasoning, we leverage open scientific video websites to extract key frames from videos that illustrate scientific phenomena, and generate textual explanation about the state transitions between extracted frames. (c) For more generalized and commonsense-level reasoning featuring daily-life scenarios, we prepare images and texts movies. Causally-related consecutive frame sequences could be extracted from movies, and textual descriptions about causality of character behaviors or transitions across the frame sequences could be formulated by collecting character dialogues and performing visual analysis on the frames.

For Micro-level reasoning data, we directly generate preliminary images via BAGEL-Think. Next we perform several rounds of self-reflection. We repetitively evaluate on generated or edited images from the previous round using VLM models such as GPT-4o, outputting an assessment of image quality and instruction-following as well as an instruction on how should the image be refined in order to align with the intention of the original prompts better. We then employ image generation models (i.e. GPT-4o) capable of image editing to edit the images in prior rounds according to the refinement instruction. We collect the textual and visual output in each self-reflection loop as the interleaved text-image data for Micro-level reasoning SFT.

**Data information.** In total, we initially collect approximately 11K complete long-form multi-modal Chain-of-Thought trajectories, which can be decomposed into 11K samples for macro-level learning and 20K samples for micro-level learning. Moreover, thanks to the decomposable nature of our framework, the model can also benefit from fragmented data. To further enhance its basic capabilities, we additionally incorporate 114K text-to-image generation samples, 68K samples from echo4o (Ye et al., 2025) dataset and 46K samples from sharegpt4o-image dataset (Chen et al., 2025b), and 46K image editing samples from sharegpt4o-image. To preserve UniCoT's multimodal understanding capabilities, we incorporate approximately 100K samples from the LLaVA-OV (Li et al., 2024) OneVision Stage training data, ensuring a balanced and evenly distributed dataset.

**Future Works for Data Collection.** While our current dataset is sufficient to refine the base unified model and elicit strong multimodal reasoning capability, its relatively small scale and predominantly synthetic composition still limit the model's ability to generalize to complex real-world scenarios.

To address this limitation, we propose a hybrid data-scaling strategy that integrates high-quality real-world data with carefully structured synthetic data. This approach enables us to expand data diversity, regulate reasoning complexity, and enhance alignment between visual transitions and textual CoT steps in a cost-effective manner (see Figure S3).

For spatial reasoning and Object manipulation, we plan to leverage large, scalable datasets from the robotics and VLA domains (e.g., Open X-Embodiment), where sequences of temporally consecutive frames naturally encode causal structure and physical interaction, and textual commands of actionable tasks assigned to the robots could be validly reformed as textual guidance between image state transitions.

For scientific or procedural reasoning, rich multimodal datasets depicting scientific phenomena, controlled experiments, and procedural workflows can be sourced from open scientific video repositories (e.g., JoVE), offering clear visual evidence of state transitions between video frames, along with off-the-shelf textual explanation of entailed natural laws or experimental procedure.

For more generalized and commonsense-level reasoning featuring daily-life scenarios, particularly those involving spatial, temporal, or causal dependencies, we design a scalable pipeline to extract causally meaningful frame sequences from publicly released movies and other long-form videos without restriction on specialized topic areas. Textual guidance or explanation about the character behaviors or transitions across frame sequences could be formulated by collecting character dialogues and performing visual analysis on the frames. This provides abundant, diverse, and realistic visual transitions to support multimodal CoT learning.

Within these data pipelines, images could be extracted directly from videos or valid from source dataset and would only require lightweight post-processing regarding quality and sizes. The GPU usage or API budgets would then mainly be contributed to the generation and the formulation of structured and meaningful textual guidance between image state transitions. As a result, these data pipelines could be feasibly scaled to the level of 100k-500k considering that images are mostly natural and would not require synthetic image generation. Together, these directions form a practical roadmap towards large-scale, high-quality, and generalizable multimodal CoT data collection.

# E   MORE EXPERIMENTS.

## E.1   RESULTS FOR IMAGE EDITING BENCHMARK

**Experiment Setup.**   For the image editing task, we benchmark Uni-CoT on GEdit-Bench (Liu et al., 2025b), RISE (Zhao et al., 2025a), and KRIS (Wu et al., 2025b). GEdit-Bench (Liu et al., 2025b) is a general benchmark that provides a diverse set of real-world editing tasks. In contrast, RISE (Zhao et al., 2025a) focuses on reasoning-informed editing across temporal, causal, spatial, and logical dimensions, while KRIS (Wu et al., 2025b) serves as a diagnostic benchmark categorizing editing tasks into factual, conceptual, and procedural knowledge types.

**Quantitative Results.**   First, we evaluate the basic editing capability of Uni-CoT on GEdit-Bench (Liu et al., 2025b). As shown in Table S3, our model achieves competitive performance compared with other approaches.

Furthermore, we report KRIS and RISE benchmark results in Table S4 and Table S5. Surprisingly, on KRIS benchmark, Uni-CoT outperforms all open-source baselines across perception, conceptual, and procedural categories. Remarkably, it also surpasses the closed-source Gemini 2.0 in overall score, highlighting its robust and interpretable editing capabilities under complex reasoning instructions. On RISE benchmark, Uni-CoT demonstrate comparable performance with Gemini 2.0 with respect to both overall performance on the four reasoning categories and the sub-dimension evaluation metrics of instruction reasoning, appearance consistency and visual plausibility.

## E.2   NANO-BANANA LIKE GENERATION

Recently, Google introduced Nano-Banana (Nano Banana, 2025), a unified image generator that extends the limits of multimodal generation. Impressively, it succeeds in tasks once considered unattainable, such as generating photorealistic landscapes from highly abstract geospatial inputs,

Table S3: Quantitative comparisons on GEdit-Bench (Liu et al., 2025b). Higher is better (↑).

| Type | Model | GEdit-Bench-EN↑ | | | GEdit-Bench-CN↑ | | |
|---|---|---|---|---|---|---|---|
| | | G_SC | G_PQ | G_O | G_SC | G_PQ | G_O |
| *Private* | Gemini 2.0 (Kampf & Brichtova, 2025) | 6.73 | 6.61 | 6.32 | 5.43 | 6.78 | 5.36 |
| | GPT-4o (Hurst et al., 2024) | *7.85* | *7.62* | *7.53* | *7.67* | *7.56* | *7.30* |
| *Open-source* | Instruct-Pix2Pix (Brooks et al., 2023) | 3.58 | 5.49 | 3.68 | - | - | - |
| | MagicBrush (Zhang et al., 2023) | 4.68 | 5.66 | 4.52 | - | - | - |
| | AnyEdit (Yu et al., 2024) | 3.18 | 5.82 | 3.21 | - | - | - |
| | OmniGen (Xiao et al., 2024) | 5.96 | 5.89 | 5.06 | - | - | - |
| | Step1X-Edit (Liu et al., 2025a) | 7.09 | 6.76 | 6.70 | 7.20 | **6.87** | 6.86 |
| | BAGEL (Deng et al., 2025) | 7.36 | **6.83** | 6.52 | 7.34 | 6.85 | 6.50 |
| | **Uni-CoT** | **7.91** | 6.24 | **6.74** | **8.01** | 6.30 | **6.87** |

Table S4: Quantitative comparisons on KRIS (Wu et al., 2025b).

| Model | Perception | | | | Conceptual Reasoning | | | Procedural Knowledge | | | Overall |
|---|---|---|---|---|---|---|---|---|---|---|---|
| | Attribute Perception | Spatial Perception | Temporal Perception | Average Score | Social Science | Natural Science | Average Score | Logical Reasoning | Instruction Decompose | Average Score | Score |
| *GPT-4o* (OpenAI) (Hurst et al., 2024) | *83.17* | *79.08* | *68.25* | *79.80* | *85.50* | *80.06* | *81.37* | *71.56* | *85.08* | *78.32* | *80.09* |
| Gemini-2.0 (Kampf & Brichtova, 2025) | 66.33 | 63.33 | 63.92 | 65.26 | 68.19 | 56.94 | 59.65 | 54.13 | 71.67 | 62.90 | 62.41 |
| Step 3φ vision (StepFun) (stepfun, 2025) | 69.67 | 61.08 | 63.25 | 66.70 | 66.88 | 60.88 | 62.32 | 49.06 | 54.92 | 51.99 | 61.43 |
| Doubao (ByteDance, 2025) | 70.92 | 59.17 | 40.58 | 63.30 | 65.50 | 61.19 | 62.23 | 47.75 | 60.58 | 54.17 | 60.70 |
| BAGEL (Deng et al., 2025) | 64.27 | 62.42 | 42.45 | 60.26 | 55.40 | 56.01 | 55.86 | 52.54 | 50.56 | 51.69 | 56.21 |
| BAGEL-Think (Deng et al., 2025) | 67.42 | 68.33 | 58.67 | 66.18 | 63.55 | 61.40 | 61.92 | 48.12 | 50.22 | 49.02 | 60.18 |
| **Uni-CoT** | **72.76** | **72.87** | **67.10** | **71.85** | **70.81** | **66.00** | **67.16** | 53.43 | **73.93** | **63.68** | **68.00** |

including satellite imagery and isohypse line maps (top of Figure S4). These outcomes shows that Nano-Banana exhibits a high-level implicit reasoning mechanism to bridge modalities with substantial domain gaps, rather than merely memorizing visual correspondences.

We target reproducing this success by introducing a reasoning-driven generation process that transforms implicit reasoning mappings into explicit and interpretable sub-tasks. Concretely, we formulate the transformation from an isohypse map to a natural landscape image as three sequential steps:

- *2d-to-3d*, interpolating the 2D contour lines into a continuous 3D terrain representation;

- *3d-crop*, projecting a plausible camera view from the global 3D terrain;

- *3d-to-real*, rendering the localized 3D view into a photorealistic landscape.

This decomposition mirrors how human experts, such as geographers or cartographers, would mentally reconstruct terrain from contour lines before imagining a realistic viewpoint.

To validate this hypothesis, we curate a dataset of 3K multi-modal Chain-of-Thought (CoT) geography samples based on GeoPose3K (Brejcha & Čadík, 2017) explicitly aligned with this decomposition. We then refine our model on this dataset with around 12,000 steps. As shown in the middle row of Figure S4, our model present a strong generation ability on this specialized test set . By contrast, when the model is fine-tuned directly on raw isohypse–landscape pairs without reasoning decomposition, training becomes unstable and the generations exhibit incoherent or distorted structures (bottom row of Figure S4).

Furthermore, we analyze the training dynamics by visualizing the mean squared error (MSE) loss curves of Uni-CoT (with explicit reasoning) versus Uni-CoT w/o Think (direct fine-tuning). As shown in Figure S5a, direct training starts with a substantially higher initial loss, reflecting the large domain gap between symbolic contours and naturalistic landscapes. More critically, while multi-modal CoT converges smoothly within ∼2k iterations, direct fine-tuning shows no stable convergence trend even after 12k iterations. This divergence underscores the intrinsic difficulty of directly mapping from 2D contour modalities to realistic 3D landscape imagery. By contrast, introducing

Table S5: Quantitative comparisons on RISE (Zhao et al., 2025a).

| Model | Temporal | Causal | Spatial | Logical | Overall |
|---|---|---|---|---|---|
| *GPT-4o* (Hurst et al., 2024) | *34.1* | *32.2* | *37.0* | *10.6* | *28.9* |
| Gemini-2.0 (Kampf & Brichtova, 2025) | 8.2 | 15.5 | **23.0** | **4.7** | **13.3** |
| BAGEL-Think (Deng et al., 2025) | 5.9 | 17.8 | 21.0 | 1.2 | 11.9 |
| BAGEL (Deng et al., 2025) | 2.4 | 5.6 | 14.0 | 1.2 | 6.1 |
| **Uni-CoT** | 8.2 | **18.9** | 20.0 | 1.2 | 12.5 |

structured reasoning decomposition not only accelerates convergence but also improves robustness, highlighting the effectiveness of reasoning-driven generation in bridging heterogeneous modalities.

### E.3 ABLATION STUDY

**Ablation of Reasoning Component.** We isolate the contribution of each reasoning component via three baselines: (1) **w/o CoT**: only with simple text reasoning; (2) **w/o macro-CoT**: only micro-level self-reflection; (3) **w/o micro-CoT**: only macro-level decomposition without iterative refinement. As shown in Table S6 left, on GenEval (Ghosh et al., 2023), macro-CoT is substantially more critical than micro-CoT. This is likely due to the nature of GenEval prompts: they are programmatically generated and often deviate from natural human phrasing. Macro-level decomposition helps bridge this gap by progressively transferring these synthetic prompts into more natural forms, thereby facilitating smoother generation. In contrast, as shown in Table S6 right, on WISE (Niu et al., 2025), micro-CoT contributes to performance substantially more than macro-CoT. We hypothesize that this arises from the nature of WISE prompts: they are abstract yet relatively simple, and therefore benefit more from fine-grained reasoning (micro-CoT) rather than explicit high-level decomposition (macro-CoT).

**Ablation on Training Strategy.** We further examine whether the proposed hierarchical training paradigm can simplify and stabilize multimodal optimization. To this end, we compare Uni-CoT with a naïve long-chain baseline (**Uni-CoT Raw**) that is directly fine-tuned on full interleaved trajectories. As shown in Figure S5, the loss of our method converges more than twice as fast (6k vs. 12k steps), and Table S7 indicates that it consistently outperforms Uni-CoT Raw across all categories. We note that these experiments were conducted under limited computational resources, and therefore the Uni-CoT results reported in Table S7 do not reflect the final performance of our full model.

Table S6: Ablation of CoT on GenEval and WISE.

| Model | GenEval (Ghosh et al., 2023) | | | | | | | WISE (Niu et al., 2025) | | | | | | |
|---|---|---|---|---|---|---|---|---|---|---|---|---|---|---|
| | Single | Two | Cnt. | Color | Pos. | Attr. | Overall | Cult. | Time | Space | Bio. | Phys. | Chem. | Overall |
| w/o CoT | 0.99 | 0.95 | 0.82 | 0.90 | 0.55 | 0.69 | 0.81 | 0.75 | 0.67 | 0.79 | 0.64 | 0.79 | 0.65 | 0.72 |
| w/o micro-CoT | 0.99 | **0.96** | 0.83 | 0.89 | **0.58** | **0.72** | **0.83** | 0.73 | **0.70** | **0.76** | 0.69 | 0.77 | 0.65 | 0.71 |
| w/o macro-CoT | 0.99 | 0.95 | 0.81 | 0.87 | 0.57 | 0.71 | 0.81 | 0.75 | 0.68 | 0.74 | 0.66 | 0.79 | 0.71 | 0.73 |
| **Uni-CoT** | 0.99 | **0.96** | **0.84** | **0.92** | 0.57 | 0.71 | **0.83** | **0.76** | **0.70** | **0.76** | **0.73** | **0.81** | **0.73** | **0.75** |

Table S7: Ablation of Training on WISE.

| Model | Cult. | Time | Space | Bio. | Phys. | Chem. | Overall↑ |
|---|---|---|---|---|---|---|---|
| Uni-CoT Raw | 0.73 | **0.67** | 0.75 | 0.60 | 0.75 | 0.65 | 0.70 |
| **Uni-CoT** | **0.75** | 0.66 | **0.78** | **0.70** | **0.78** | **0.71** | **0.73** |

**Ablation on Training Strategy.** We examine the training dynamics of our approach (Uni-CoT) against a naïve baseline (Uni-CoT Raw) that directly fine-tunes on long-chain interleaved data. As shown in Figure S5b, our method converges substantially faster, reaching a comparable loss level within only 6,000 steps, whereas the baseline requires 12,000 steps. When both models are trained for 12,000 steps on a dataset of ∼10,000 samples, we further evaluate them on WISE (Table S7). Across all metrics, our method consistently surpasses the long-chain baseline, demonstrating higher training efficiency and superior output quality.

**Ablation study for base model**   To further assess the generality of our Uni-CoT framework, we replace its original base model, Bagel, with BLIP3o (Chen et al., 2025a). BLIP3o is a unified multimodal model built on the autoregressive backbone Qwen2.5-VL, supporting both image understanding and image generation. For understanding, it encodes visual inputs using a CLIP encoder and predicts text tokens via standard cross-entropy training. For generation, the model first produces intermediate visual features in a flow-style manner, which then condition a diffusion transformer to synthesize images. Through this unified design, BLIP3o aligns visual perception and visual generation within a single coherent architecture.

We evaluate this setup on the WISE benchmark and investigate the effectiveness of a key Uni-CoT reasoning component, the micro-CoT self-reflection mechanism. Specifically, we fine-tune the BLIP3o (the BLIP3o-NEXT-edit-VAE (BLIP3o Team, 2025) version for its competitive image editing capability) on our training dataset using 8×A100 GPUs for 100,000 steps, which took approximately 16 hours. We then compare three baselines to analyze the contribution of self-reflection: (1) BLIP3o: the original pretrained model without any fine-tuning; (2) BLIP3o-sft w/o Uni-CoT: the model fine-tuned on our dataset *without* self-reflection signals; (3) BLIP3o-sft w/ Uni-CoT: the model fine-tuned on our dataset *with* micro-CoT self-reflection.

As shown in Table S8, directly fine-tuning BLIP3o on our dataset already yields a substantial performance improvement. This is likely because the selected BLIP3o version (BLIP3o-NEXT-edit-VAE) exhibits relatively weaker reasoning alignment in its original form. Furthermore, with our reasoning architecture incorporated, BLIP3o-SFT w/ Uni-CoT achieves an additional improvement of approximately 0.8 points across all metrics compared to BLIP3o-SFT w/o Uni-CoT. This clearly demonstrates the effectiveness of our proposed multimodal reasoning strategy. The visualization of reasoning trajectory of BLIP3o is shown in Figure S9.

Table S8: Ablation study for base model

| Model | Culture | Time | Space | Biology | Physics | Chemistry | Overall↑ |
|---|---|---|---|---|---|---|---|
| BLIP3o | 0.30 | 0.33 | 0.48 | 0.37 | 0.36 | 0.22 | 0.33 |
| BLIP3o-sft w/o Uni-CoT | 0.54 | 0.54 | 0.59 | 0.42 | 0.61 | 0.57 | 0.55 |
| **BLIP3o-sft w/ Uni-CoT** | **0.65** | **0.62** | **0.67** | **0.48** | **0.70** | **0.69** | **0.64** |

# F   MORE VISUALIZATION

We present the image editing visualization results in Figure S6. Additional image editing–based geography reasoning cases are provided in Figure S7, while further examples of macro-CoT, micro-CoT, and hybrid macro–micro CoT are illustrated in Figure S8.

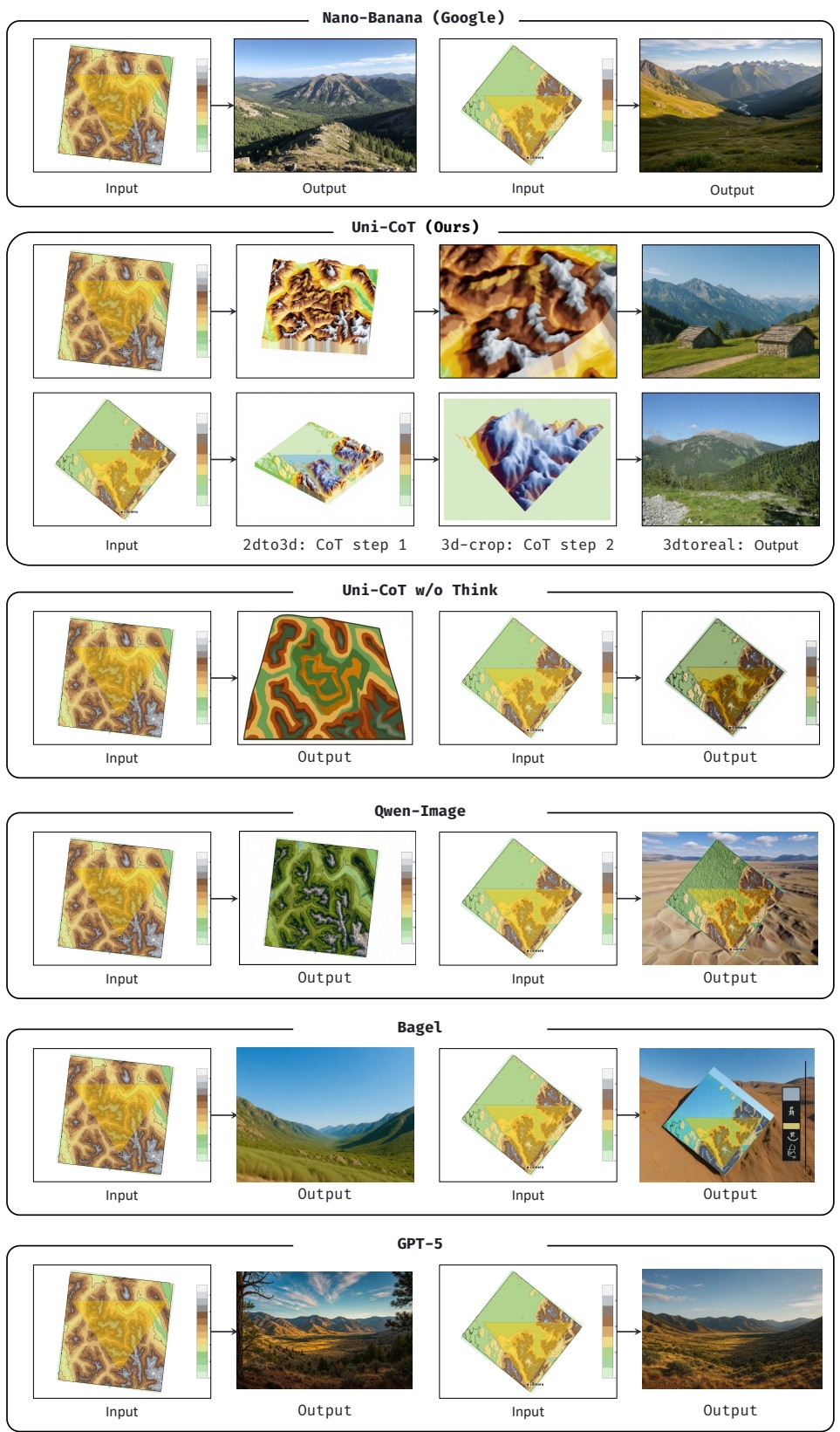

Figure S4: Qualitative Results for Nano-Banana like Generation.

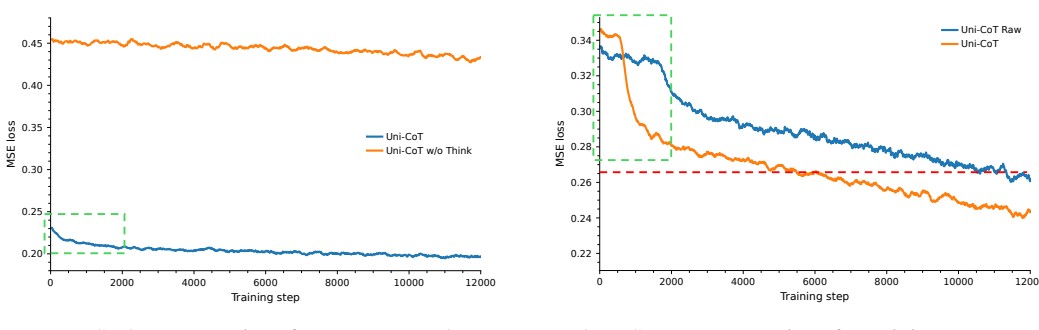

(a) MSE loss comparison for geometry task.

(b) MSE Loss comparison for training strategy.

Figure S5: MSE loss comparisons.

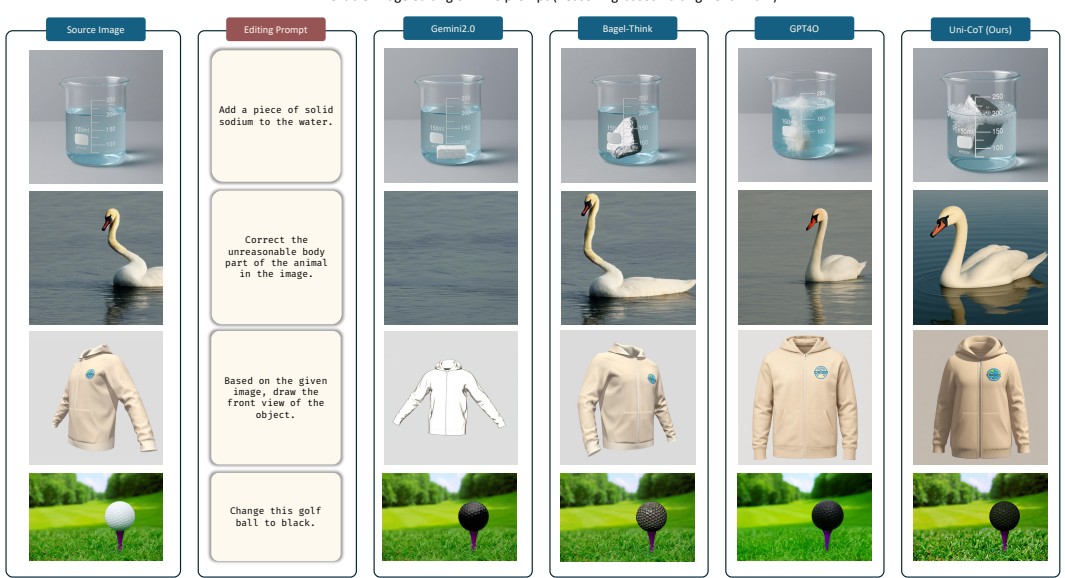

Figure S6: Qualitative Results for Reliable Image Editing. Uni-CoT demonstrates considerable image editing abilities, further supporting the effectiveness of its micro-level CoT reasoning. It can generate textual editing instructions and modify the current visual state accordingly.

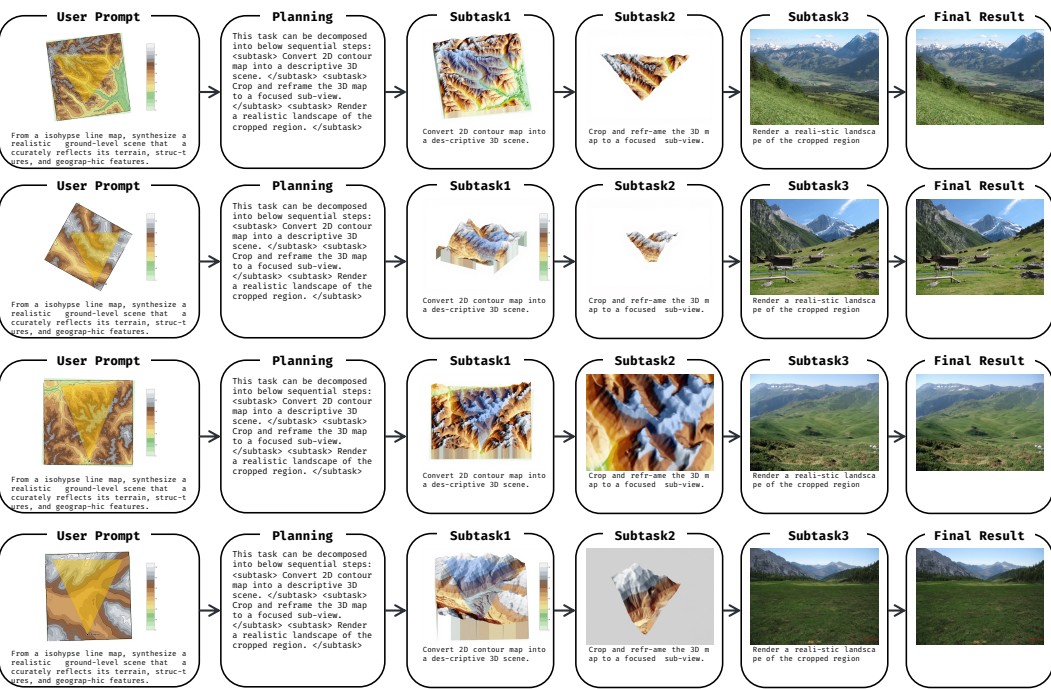

Figure S7: More visualization results for Geography Reasoning.

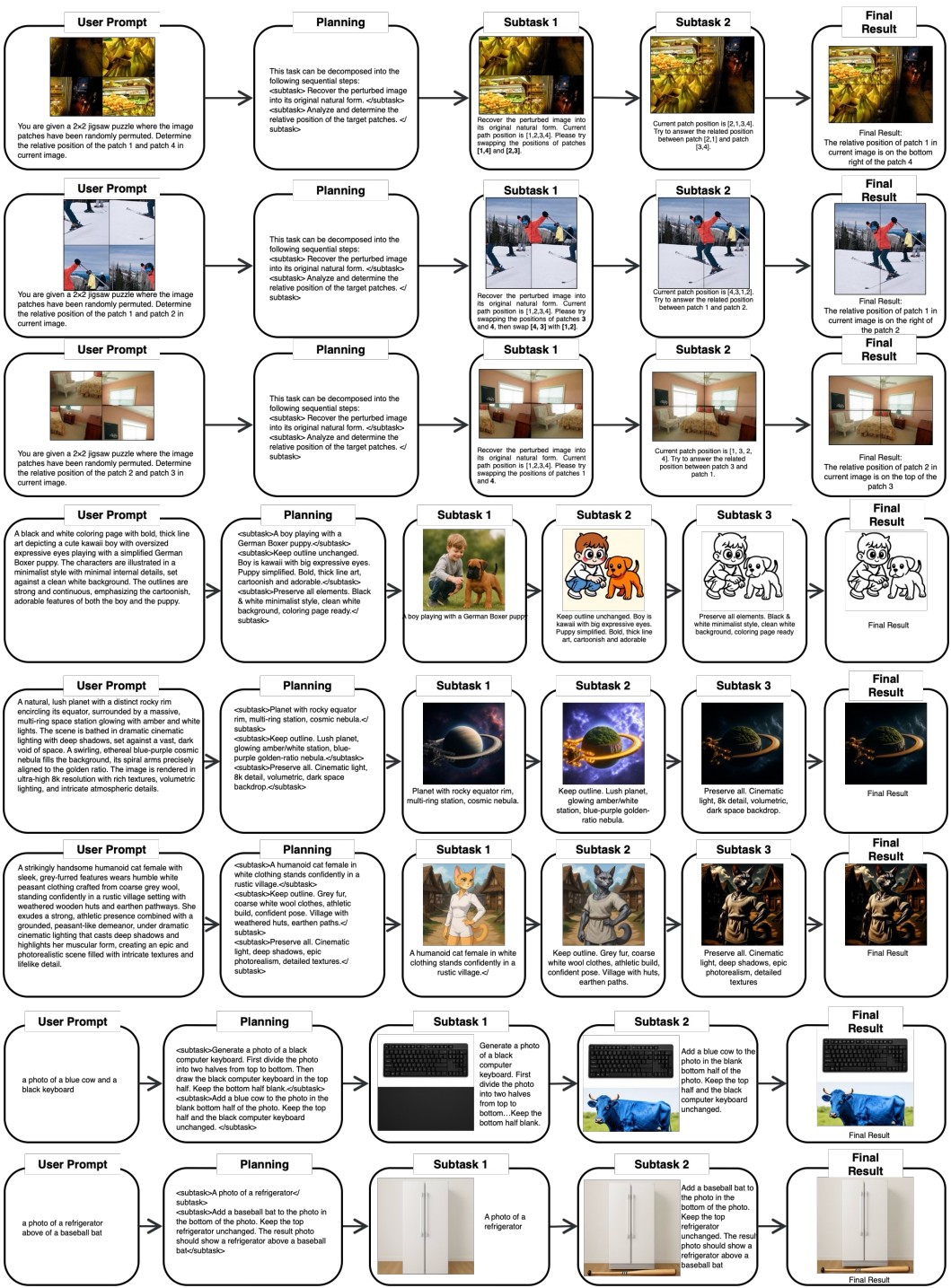

Figure S8: More Visualization of our Uni-CoT thinking process.

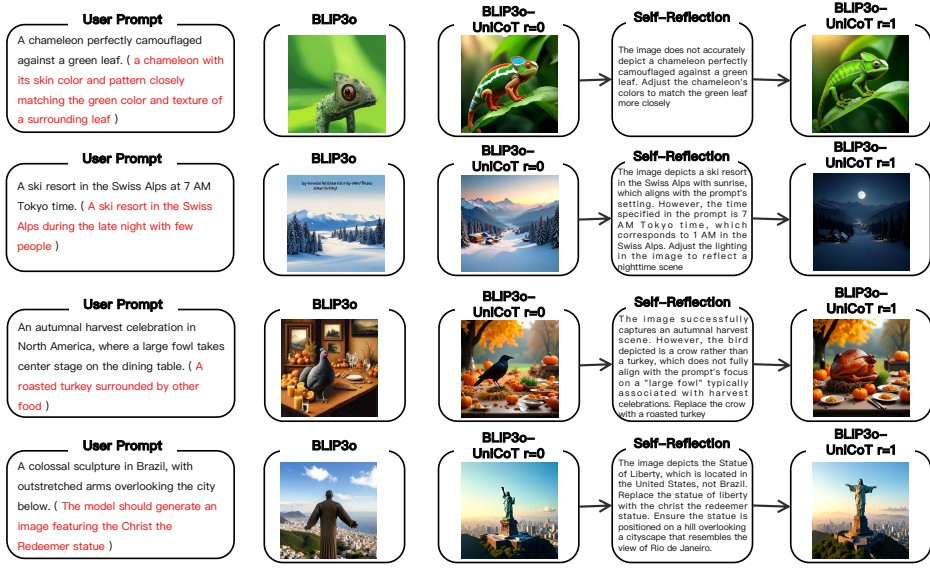

Figure S9: More Visualization of our Uni-CoT with BLIP3o model's thinking process.

