# OpenReview forum: "Uni-CoT: Towards Unified Chain-of-Thought Reasoning Across Text and Vision"
_ICLR.cc/2026/Conference — ICLR 2026 Poster_

### Official Review · Reviewer_XhV7 · 2025-10-29

**Soundness:** 4
**Presentation:** 3
**Contribution:** 2
**Rating:** 6
**Confidence:** 2

**Summary:**

This paper proposes Uni-CoT, a Unified Chain-of-Thought framework that captures structured visual transitions and seamlessly aligns them with textual logic. They use a macro-level CoT for high-level planning and a micro-level CoT for localized subtask execution.  Overall, Uni-CoT performs better than the baselines.

**Strengths:**

1. There are still relatively few reasoning frameworks for unified models, and this study effectively fills that gap.

2. The paper is easy to follow, the figures are very clear, and the code is open-sourced.

3. The experiments are highly detailed, and the provided examples are intuitive and illustrative.

4. The selected datasets are convincing (e.g., WISE), and the experimental results are strong.

**Weaknesses:**

Although the paper has several notable strengths, its weaknesses are also apparent. The proposed method does not appear sufficiently novel since it builds a reasoning framework using prompting-based scaffolding, an approach that has already been extensively and systematically explored in the LLM and VLM literature. Therefore, the method itself is not particularly innovative. Nevertheless, this paper is likely among the first to adapt such approaches to unified models, which remains a meaningful contribution.

**Questions:**

I am not very familiar with the literature on unified models, but I would like to ask why there are missing data entries in Tables 1 and 3. Is this due to limited computational resources that made it difficult to obtain the results, or are there other reasons?

---

> ### Author Response · Authors · 2025-11-19
> **Reviewer XhV7— Response [1/2]**
>
> We sincerely thank the reviewer for the thoughtful and constructive comments. In the following sections we address these points in detail.
> We have also revised our paper according to the reviewers’ suggestions. In the updated manuscript, newly added content is highlighted in blue, and sections to be deleted or relocated are marked in green.
>
>
> **1. Although the paper has several notable strengths, its weaknesses are also apparent. The proposed method does not appear sufficiently novel since it builds a reasoning framework using prompting-based scaffolding, an approach that has already been extensively and systematically explored in the LLM and VLM literature. Therefore, the method itself is not particularly innovative. Nevertheless, this paper is likely among the first to adapt such approaches to unified models, which remains a meaningful contribution.**
>
> Thanks for the thoughtful comment. We agree that prompting-based scaffolding has been extensively studied in both LLMs and VLMs, and we appreciate that the reviewer's noting about our adaptation to unified models is meaningful. We would like to clarify, however, that *Uni-CoT is not a prompting-based or training-free scaffolding method*. Instead, it is a *supervised, model-internal hierarchical reasoning framework* designed specifically to address limitations of unified models.
>
> To clearly articulate our contributions relative to prior scaffolding-based work, we summarize the key points below:
>
> 1. **A unified cross-modal CoT framework implemented *within* a single model.**
>    Unlike prompting-based scaffolds or multi-agent pipelines that operate outside the model, Uni-CoT integrates both macro- and micro-level reasoning *inside* the unified architecture. This enables end-to-end multimodal CoT that jointly handles text and image reasoning without external controllers.
>
> 2. **A new observation on the core bottleneck of multimodal CoT.**
>    We identify that unified models struggle with interleaved reasoning primarily because each reasoning step requires processing of thousands of visual tokens, leading to prohibitive quadratic attention costs. This observation motivates a decomposed trajectory design and explains why existing unified models, despite rapid progress, still lack practical multimodal CoT capabilities.
>
> 3. **A hierarchical structure explicitly designed to reduce complexity while preserving reasoning behavior.**
>    Our macro–micro decomposition and MDP-style micro-CoT are not generic efficient-attention tricks. They are tailored to maintain multi-step reasoning fidelity while reducing visual-token interactions from quadratic to near-linear. This reasoning-preserving efficiency design differs fundamentally from prior efficient-attention methods that do not target multi-step multimodal reasoning.
>
> We hope that this better clarifies how Uni-CoT differs from prior scaffolding approaches and where its novelty lies.

---

> > ### Comment · Reviewer_XhV7 · 2025-11-19
> >
> > I'm satisfied with this explanation.

---

> ### Author Response · Authors · 2025-11-19
> **Reviewer XhV7— Response [2/2]**
>
> **2. I am not very familiar with the literature on unified models, but I would like to ask why there are missing data entries in Tables 1 and 3. Is this due to limited computational resources that made it difficult to obtain the results, or are there other reasons?**
>
> Thank you for pointing this out. The missing entries in Tables 1 and 3 arise because several baselines are reported directly from prior works or from the original benchmark leaderboards. Due to limited time and computational resources, we were unable to re-run all models ourselves.
>
> Therefore, the score entries in the tables are filled by following either the official benchmark leaderboards offered by the benchmark providers or the models that evaluated on the benchmarks and report their scores. For Table 1, we follow the standardized GenEval scores reported by the corresponding models listed in the table, following the official evaluation protocol.(i.e. Flux [1], Janus-Pro-7B [2], Bagel [3], etc.) For the general understanding benchmarks in Table 3, we follow either the standardized scores reported by the benchmark leaderboards (i.e. MMMU [4], etc.) or the results reported by the corresponding models (i.e. Qwen2.5-VL-7B [5], internVL-2.5B [6], etc.). For Jigsaw-R1, we follow the scores reported by Jigsaw-R1 [7].
>
> Importantly, to ensure that our core claims are not affected by measurement inconsistencies, we re-evaluated our base model (BAGEL [3]) on every benchmark and report its accurate, real-time performance under the same evaluation setup used for Uni-CoT. By grounding all comparisons against a freshly measured baseline, we minimize discrepancies caused by evaluation noise or reporting variations in prior works. Fortunately, except for the GenEval benchmark, Bagel’s results are consistent with their official numbers.
>
> [1] Black Forest Labs. Flux, 2024. URL https://github.com/black-forest-labs/flux.
>
> [2] Chen, Xiaokang, et al. "Janus-pro: Unified multimodal understanding and generation with data and model scaling." arXiv preprint arXiv:2501.17811 (2025).
>
> [3] Deng, Chaorui, et al. "Emerging properties in unified multimodal pretraining." arXiv preprint arXiv:2505.14683 (2025).
>
> [4] Yue, Xiang, et al. "Mmmu: A massive multi-discipline multimodal understanding and reasoning benchmark for expert agi." Proceedings of the IEEE/CVF Conference on Computer Vision and Pattern Recognition. 2024.
>
> [5] Bai, Shuai, et al. "Qwen2. 5-vl technical report." arXiv preprint arXiv:2502.13923 (2025).
>
> [6] Chen, Zhe, et al. "Expanding performance boundaries of open-source multimodal models with model, data, and test-time scaling." arXiv preprint arXiv:2412.05271 (2024).
>
> [7] Wang, Zifu, et al. "Jigsaw-R1: A Study of Rule-based Visual Reinforcement Learning with Jigsaw Puzzles." arXiv preprint arXiv:2505.23590 (2025).

---

### Official Review · Reviewer_GGF1 · 2025-10-31

**Soundness:** 3
**Presentation:** 2
**Contribution:** 2
**Rating:** 4
**Confidence:** 3

**Summary:**

This paper proposes Uni-CoT, a unified hierarchical Chain-of-Thought (CoT) framework for multimodal reasoning. The approach aims to bridge textual and visual reasoning by introducing two levels of CoT:

1. Macro-CoT, which handles high-level planning and decomposition of a multimodal task into subgoals.

2. Micro-CoT, which focuses on localized subtask execution, reasoning over both text and image modalities with shorter context windows.

**Strengths:**

Strengths

1. The idea of hierarchical reasoning decomposition (macro for planning, micro for execution) is conceptually sound and aligns with cognitive and hierarchical RL principles.

2. The insight that only the latest local state is needed for Micro-CoT, rather than the entire reasoning chain, is practical and can indeed reduce context length and compute cost.

3. The paper is well-motivated by the challenges of multimodal reasoning, and the integration of CoT-style reasoning into a unified visual–text model is timely

**Weaknesses:**

1. Hierarchical decomposition (macro/micro planning) has been explored in multiple contexts, including text-only CoT, hierarchical RL, and multi-agent reasoning. The paper does not sufficiently articulate what is new in Uni-CoT beyond combining these known techniques.

2. The paper omits critical information about the base model, data scale, loss functions, and training setup, and most are deferred to the appendix. This makes it difficult to assess reproducibility and fairness of comparisons.

3. In Tables 1 and 2, Uni-CoT shows only modest gains over plain CoT baselines (especially on WISE). The discussion around why and when Uni-CoT helps is minimal.

4. The ablation studies do not isolate the effects of the macro vs. micro components, nor do they show what happens when hierarchical supervision is removed.

5. Complexity reduction claim is unconvincing. The claimed reduction from quadratic to near-linear attention complexity seems theoretical. Given that the SFT data only involves 2–3 subtasks, it’s unclear that Uni-CoT’s decomposition meaningfully reduces overall computation compared to a single CoT pass.

6. Qualitative results lack rigor. For instance, in Fig. 3 (line 440), the generated image contradicts the textual description (“sealed container” appears open), suggesting incomplete alignment between reasoning and generation.

**Questions:**

1. What is the base multimodal model used?

2. Was SFT with plain CoT data attempted as a control baseline?

3. How much does each level (macro/micro) contribute individually?

4. Can the authors provide concrete statistics or examples showing context length reduction and training efficiency gains?

5. How are the auxiliary tasks (e.g., reward estimation) implemented? are they learned jointly or pre-trained separately?

---

> ### Author Response · Authors · 2025-11-19
> **Reviewer GGF1— Response [1/4]**
>
> We sincerely thank the reviewer for the thoughtful and constructive comments. In the following sections we address each point in detail.
> We have also revised our paper according to the reviewers’ suggestions. In the updated manuscript, the revised content is highlighted in blue, and sections to be deleted or relocated are marked in green.
>
> **1. Hierarchical decomposition (macro/micro planning) has been explored in multiple contexts, including text-only CoT, hierarchical RL, and multi-agent reasoning. The paper does not sufficiently articulate what is new in Uni-CoT beyond combining these known techniques.**
>
> We agree that in our initial submission, we did not sufficiently differentiate Uni-CoT from prior frameworks such as hierarchical CoT, hierarchical RL, and multi-agent frameworks. We have now revised the related-work section to address this (page 10, line 509–517).
>
> We also clarify that **Uni-CoT is not a simple combination of existing techniques**. Instead, it tackles challenges *unique to multimodal CoT reasoning* and introduces several new ideas to the community:
>
> 1. **A new observation on the bottleneck of multimodal CoT.**
>    We have identified a key factor why current unified models struggle with interleaved multimodal reasoning: the substantial computational cost incurred by repeatedly processing visual tokens throughout long reasoning trajectories. This bottleneck helps explain why, despite several unified models released over the past year [1,2], progress on interleaved multimodal CoT within a single model has remained limited.
>
> 2. **A new perspective on applying hierarchical design to reduce complexity, including a decomposable training and inference paradigm.**
>     Although hierarchical decomposition has been explored in frameworks such as text-only CoT, hierarchical RL, and multi-agent reasoning, prior work primarily focuses on improving reasoning quality, controllability, or planning efficiency. In contrast, our work adopts a fundamentally different perspective when building our hierarchical architecture. One of the core challenges of unified multimodal CoT models is to reduce the substantial computational cost introduced by repeat processing of visual tokens, meanwhile still preserving strong reasoning ability. To address this challenge, we leverage hierarchical framework design.
>
>     To this end, we introduce a planning-based macro-CoT and a self-reflection–based micro-CoT, which together decompose long multimodal reasoning trajectories into smaller, more manageable segments. Building on this hierarchical structure, we further propose a decomposable training framework that simplifies and stabilizes multimodal CoT learning, as well as a hierarchical inference procedure that accelerates interleaved multimodal reasoning. This hierarchical design for complexity reduction in unified multimodal CoT is, to our knowledge, new to the community.
>
> 3. **A new MDP-based formulation of self-reflection reasoning.**
>    We further develop a Markov Decision Process–based view of self-reflection. Leveraging the fact that self-reflection only depends on the current state, we decompose the reasoning process into a **Markov chain**, substantially simplifying training and inference while preserving the effectiveness of iterative refinement. Although recent text-only works have touched on MDP-style CoT, our formulation is developed independently and differs fundamentally: our design is tailored to multimodal reasoning and is tightly coupled with the hierarchical decomposition above.
>
> In summary, Uni-CoT’s novelty lies in **identifying the complexity bottleneck of multimodal CoT** and proposing a **systematic solution**. The solution includes a hierarchical training/inference paradigm and an MDP-based self-reflection mechanism, addressing the computation complexity bottlenet effectively within a unified model.
>
> [1] Xie, Jinheng, et al. "Show-o: One single transformer to unify multimodal understanding and generation." arXiv preprint arXiv:2408.12528 (2024).
>
> [2] Chen, Xiaokang, et al. "Janus-pro: Unified multimodal understanding and generation with data and model scaling." arXiv preprint arXiv:2501.17811 (2025).

---

> ### Author Response · Authors · 2025-11-19
> **Reviewer GGF1— Response [2/4]**
>
> **2. The paper omits critical information about the base model, data scale, loss functions, and training setup, and most are deferred to the appendix. This makes it difficult to assess reproducibility and fairness of comparisons.**
>
> We apologize for not clearly presenting several key implementation details in the main paper, as many were originally placed in Appendix C. In the revised version (page 7, line 333-340), we have moved these essential details into the main text and added clear guidance on where to find supplementary information in the appendix.
>
> Specifically, Uni-CoT is built on top of the unified model Bagel [1] and is finetuned on our 31k-sample multimodal CoT dataset. Training is supervised using cross-entropy loss for textual reasoning and mean-squared error loss for visual reasoning. We also describe the learnig rate, token packing strategy, and GPU usage in the main paper.
>
> To further support reproducibility and ensure fairness, we have provided code for preview via an anonymous link https://anonymous.4open.science/r/UniCoT/ (as noted in the abstract). We will release additional implementation detail, including the full dataset and complete training pipeline, in subsequent updates.
>
> [1] Deng, Chaorui, et al. "Emerging properties in unified multimodal pretraining." arXiv preprint arXiv:2505.14683 (2025).
>
> **3. In Tables 1 and 2, Uni-CoT shows only modest gains over plain CoT baselines (especially on WISE). The discussion around why and when Uni-CoT helps is minimal.**
>
> Thanks for raising this point. Regarding performance, we note that both GenEval and WISE are highly challenging multimodal reasoning benchmarks, for which large performance gains are inherently difficult to obtain. Despite that, several reviewers (JwV8: “strong results”; MA39 and xg6H: “SOTA performance”; XhV7: “results are strong on WISE”) recognized that Uni-CoT provides consistent and meaningful improvements.
>
> A finer-grained analysis further shows that these gains are significant. On WISE, Uni-CoT achieves substantial improvements on science-related tasks: it outperforms the plain CoT baselines [Bagel-think or Uni-CoT w/o CoT] by more than 0.08 on Biology and Chemistry and even surpasses GPT-4o on Physics. These results demonstrate that Uni-CoT is particularly effective for tasks requiring structured scientific reasoning.
>
> To clarify why and when Uni-CoT helps, we have expanded the ablation discussion in the main paper (page 9, line 448-466) for illustration:
>
> On GenEval, the left part of Table 4 (page 9, line 441-445) shows that macro-level CoT is the dominant contributor. We believe this is because the prompt of Geneval is generated by code, which usually is unatural and mostly counter-intertuitive. Meanwhile, macro-level CoT can reorganize these unnatural prompts into a sequence of coherent and natural subtasks, effectively bridging the gap between synthetic prompt contents and real-world generation patterns. This highlights that when a task requires structural reorganization of the prompt, macro-level reasoning becomes particularly beneficial.
>
> In contrast, our ablation on WISE (the right part of Table 4, page 9, line 441-445) shows that the improvements come mainly from micro-level self-reflection. WISE prompts are concise, human-written, and already well-structured, so macro-level decomposition provides limited gain. However, WISE often requires strong, fine-grained reasoning, which benefits more from iterative step-by-step refinement. Thus, micro-level self-reflection becomes the key contributor on WISE by enabling precise reasoning transitions without altering the original prompt structure.
>
> In summary, Uni-CoT is most effective when:
> (1) the prompt requires structural reorganization (i.e. on GenEval), or
> (2) the task requires fine-grained semantic reasoning (i.e. in science domains such as on WISE).
>
> **4. The ablation studies do not isolate the effects of the macro vs. micro components, nor do they show what happens when hierarchical supervision is removed.**
>
> We agree that clearer ablations are needed. The macro–vs.–micro and hierarchical–vs.–single-chain ablations were originally in the Appendix E3 (Tables S6–S8), and have now been moved to the main paper (page 9, lines 448–466).
>
> The ablation results are summarized in the table below with more detail presented in the main paper (Table 4 and 5, page 9, line 441-445). From these results, we observe the following:
>
> * **GenEval:** Removing **macro-CoT** reduces performance from **0.83 to 0.81**, while removing **micro-CoT** preserves the full score (**0.83**). This indicates that **macro-level decomposition is the primary contributor** on GenEval.
>
> * **WISE:** Removing **micro-CoT** decreases performance from **0.75 to 0.71**, while removing **macro-CoT** causes only a minor drop from **0.75 to 0.73**. This shows that **micro-level self-reflection is the primary contributor** on WISE.

---

> ### Author Response · Authors · 2025-11-19
> **Reviewer GGF1— Response [3/4]**
>
> **Table: Ablation on GenEval and WISE**
> | **Model**             | **GenEval ↑** | **WISE ↑** |
> | --------------------- | ------------: | ---------: |
> | Uni-CoT w/o CoT       |          0.81 |       0.72 |
> | Uni-CoT w/o micro-CoT |      **0.83** |       0.71 |
> | Uni-CoT w/o macro-CoT |          0.81 |       0.73 |
> | **Uni-CoT**           |      **0.83** |   **0.75** |
>
> **5. Complexity reduction claim is unconvincing. The claimed reduction from quadratic to near-linear attention complexity seems theoretical. Given that the SFT data only involves 2–3 subtasks, it’s unclear that Uni-CoT’s decomposition meaningfully reduces overall computation compared to a single CoT pass.**
>
> We heartily appreciate the reviewer’s suggestion for a more rigorous analysis of complexity reduction. In response, we quantify the average token interaction counts under (a) a single-pass CoT (Uni-CoT Raw) and (b) Uni-CoT’s hierarchical reasoning framework (Uni-CoT) on 100 sampled prompts.
> As can be concluded from the results, the average token interaction is nearly quadratic when using a single CoT pass, while be near-linear when employing the decomposiiton strategy in Uni-CoT. Empirically, on our SFT setting with 2–3 subtasks, Uni-CoT reduces the average token interaction count by factors of 2.24× (two subtasks) and 2.95× (three subtasks) relative to a single CoT pass, the gap between Uni-CoT and a single CoT pass widens as the subtask count grows.
> We thank the reivewer again for raising such a beneficial question,
> this quantitative results would further clarify the ncecessity and effectiveness of the decomposition strategy used by Uni-CoT.
>
> | Reasoning Step Num | Uni-CoT  | Uni-CoT Raw |
> |--------------------|----------|-------------|
> | 0                  | 3.56e+07 | 8.45e+07    |
> | 1                  | 7.63e+07 | 1.71e+08    |
> | 2                  | 1.17e+08 | 3.46e+08    |
> | 3                  | 1.57e+08 | 6.11e+08    |
> | 4                  | 1.98e+08 | 9.68e+08    |
> | 5                  | 2.39e+08 | 1.41e+09    |
> | 6                  | 2.79e+08 | 1.95e+09    |
> | 7                  | 3.20e+08 | 2.57e+09    |
> | 8                  | 3.60e+08 | 3.28e+09    |
> | 9                  | 4.01e+08 | 4.09e+09    |
> | 10                 | 4.42e+08 | 4.98e+09    |
>
> **6. Qualitative results lack rigor. For instance, in Fig. 3 (line 440), the generated image contradicts the textual description (“sealed container” appears open), suggesting incomplete alignment between reasoning and generation.**
>
> Thank you for pointing this out. We acknowledge that in Figure 3, the “sealed container” example remains imperfect in the final output. This reflects a limitation of unified models, the inability to reliably detect certain fine-grained semantic inconsistencies, despite our micro-level self-reflection. Based on this beneficial review, we have added a brief discussion on this limitation in the revised manuscript (page 10, line 536-539).
>
> At the same time, we would like to invite the reviewer to consider other visualizations in our results (e.g., Fig. S7, page 28 and Fig. S8, page 29) . These examples more clearly demonstrate that, despite occasional failures, Uni-CoT consistently produces coherent reasoning trajectories and semantically aligned generations. Overall, while limitations remain, our framework delivers substantial improvements over BAGEL in both reasoning quality and visual correctness.
>
> ### Questions:
>
> **1. What is the base multimodal model used?**
>
> Uni-CoT is built on BAGEL [1], a unified multimodal model that integrates SigLIP2-based ViT for image understanding and FLUX-based VAE for image generation. A brief overview of BAGEL is provided in Section 2.
>
> We are sorry about not describing in full detail about our base model in the implementation details. In the revised version, we have added a clear and complete description of the base model to the Implementation Detail section (page 7 line 333-340).
>
> [1] Deng, Chaorui, et al. "Emerging properties in unified multimodal pretraining." arXiv preprint arXiv:2505.14683 (2025).
>
> **2. Was SFT with plain CoT data attempted as a control baseline?**
>
> Yes. In our initial submission we have included controls to identify the effectivness of our design (in the page 9, line 441-445):
>
> - Uni-CoT with raw CoT: SFT on full-length multi-modal CoT sequences without hierarchical supervision.
>
> - Uni-CoT: SFT on hierarchical multi-modal CoT sequences
>
> As shown in below, both Uni-CoT without CoT and Uni-CoT with raw CoT show inferior performance compared to Uni-CoT. We note that these experiments were conducted under limited computational resources, and therefore the Uni-CoT results reported in below Table do not reflect
> the final performance of our full model.

---

> ### Author Response · Authors · 2025-11-19
> **Reviewer GGF1— Response [4/4]**
>
> | **Model**      | **Cult.** | **Time** | **Space** | **Bio.** | **Phys.** | **Chem.** | **Overall ↑** |
> |----------------|-----------|----------|-----------|----------|-----------|-----------|----------------|
> | Uni-CoT Raw    | 0.73      | **0.67** | 0.75      | 0.60     | 0.75      | 0.65      | 0.70           |
> | **Uni-CoT**    | **0.75**  | 0.66     | **0.78**  | **0.70** | **0.78**  | **0.71**  | **0.73**       |
>
>
> **3. How much does each level (macro/micro) contribute individually?**
>
> As shown below, the contributions of macro- and micro-CoT differ across benchmarks (more details refer to main paper, page 9, line 446-466):
>
> * **GenEval:** Removing **macro-CoT** reduces performance from **0.83 to 0.81**, while removing **micro-CoT** preserves the full score (**0.83**). This indicates that the main contribution on GenEval comes from **macro-level decomposition**.
>
> * **WISE:** Removing **micro-CoT** decreases performance from **0.75 to 0.71**, whereas removing **macro-CoT** causes only a slight drop from **0.75 to 0.73**. This shows that the dominant factor on WISE is **micro-level self-reflection**.
>
>
> **Table: Ablation on GenEval and WISE**
> | **Model**             | **GenEval ↑** | **WISE ↑** |
> | --------------------- | ------------: | ---------: |
> | Uni-CoT w/o CoT       |          0.81 |       0.72 |
> | Uni-CoT w/o micro-CoT |      **0.83** |       0.71 |
> | Uni-CoT w/o macro-CoT |          0.81 |       0.73 |
> | **Uni-CoT**           |      **0.83** |   **0.75** |
>
> **4. Can the authors provide concrete statistics or examples showing context length reduction and training efficiency gains?**
>
> As for the context length reduction, we report the average token interaction involved in Uni-CoT and Uni-CoT Raw( Uni-CoT model without the proposed hierarchical framework). As can be concluded, Uni-CoT effectively reduce the context length that the average token interaction number is considerable less than that of the Uni-CoT Raw. Further, owing to our hierarchical framework, Uni-CoT’s average token interactions grow near-linearly with the number of reasoning steps, whereas Uni-CoT Raw exhibits quadratic growth.
>
> | Reasoning Step Num | Uni-CoT  | Uni-CoT Raw |
> |--------------------|----------|-------------|
> | 0                  | 3.56e+07 | 8.45e+07    |
> | 1                  | 7.63e+07 | 1.71e+08    |
> | 2                  | 1.17e+08 | 3.46e+08    |
> | 3                  | 1.57e+08 | 6.11e+08    |
> | 4                  | 1.98e+08 | 9.68e+08    |
> | 5                  | 2.39e+08 | 1.41e+09    |
> | 6                  | 2.79e+08 | 1.95e+09    |
> | 7                  | 3.20e+08 | 2.57e+09    |
> | 8                  | 3.60e+08 | 3.28e+09    |
> | 9                  | 4.01e+08 | 4.09e+09    |
> | 10                 | 4.42e+08 | 4.98e+09    |
>
> As for tht training efficiency gains, we ablate the proposed hierarchical training framework in the main paper, Figure 4 (page 10, line 486-496) and Table 5 (page 9, line 441-445). As can be concluded, the loss of our method not only converges more than twice as fast (6k vs. 12k steps), but also consistently outperforms Uni-CoT Raw across all categories.
>
> **5. How are the auxiliary tasks (e.g., reward estimation) implemented? are they learned jointly or pre-trained separately?**
>
> The details of all auxiliary tasks, including their loss functions and data formats—are provided in Appendix C. At the macro level, Uni-CoT is trained only with standard CE loss (for planning and textual summarization) and joint CE+MSE loss (for image synthesis).
>
> All auxiliary tasks are introduced exclusively in the micro-level CoT, where self-reflection is modeled as a Markov Decision Process. **These tasks are first pre-trained separately and then jointly learned.** The unified model shares the same parameters across all objectives and is optimized end-to-end on interleaved multi-modal traces.
>
> The four auxiliary objectives are:
> * **Text Action Generation:** Predicting a textual editing instruction; supervised via CE.
> * **Image Action Generation:** Predicting visual modifications; supervised via MSE.
> * **Next-State Prediction:** Summarizing the updated image into a new reasoning state; supervised via CE.
> * **Reward Estimation:** Predicting a textual evaluation of the intermediate result; also supervised via CE.
>
> For reward estimation specifically, the training data explicitly includes a short evaluation comment for each intermediate state (e.g., “The flame should be removed,” “The layout is still incorrect”), with **“Everything is good.”** used to denote task completion. These evaluations serve directly as the supervision signal for the reward head, implemented simply as next-token prediction over text.

---

### Official Review · Reviewer_xg6H · 2025-11-01

**Soundness:** 3
**Presentation:** 3
**Contribution:** 3
**Rating:** 6
**Confidence:** 2

**Summary:**

This paper proposes a hierarchical framework to extend chain-of-thought (CoT) reasoning to multi-modal settings. The authors introduce Uni-CoT, which unifies text and visual reasoning within a single model by integrating structural visual transitions and coherent textual logic. To manage the high computational cost of multi-modal reasoning, they design a macro–micro hierarchical CoT, where the macro level plans subgoals and the micro level executes them via Markov Decision Process–based self-reflection. Experiments on reasoning-driven image generation and understanding benchmarks show state-of-the-art results and improved interpretability, demonstrating Uni-CoT’s potential for scalable and coherent multimodal reasoning .

**Strengths:**

1. Originality: This paper proposes a unified framework for multi-modal chain-of-thought (CoT) reasoning, bridging visual and textual reasoning within a single coherent process. The Markov Decision Process formulation in this paper reduces the computation overhead.

2. Quality: The proposed method is well-motivated, combining goal decomposition with self-reflective decision-making. The experiments are comprehensive, covering both understanding and generation tasks, and the results consistently support the framework’s effectiveness.

3. Clarity: The paper is generally well-organized and easy to understand.

4. Significance: The work contributes meaningfully to the emerging area of unified multimodal reasoning. It also formally analyzes the complexity of efficient MDP attention.

**Weaknesses:**

1. The main novelty is kind of limited; there are some similar works using CoT for image generation refinement, like LayerCraft [1]. Efficient attention is also a well-studied area. However, combining them might be the first try.
2. The training dataset for CoT image generation purely depends on synthetic data. This limits generalization to open-domain or real-world visual reasoning tasks.
3. Figure S3 in the appendix seems not accurately reflect a realistic ground-level scene, which raises questions about the accuracy of synthetic training data.

[1] LayerCraft: Enhancing Text-to-Image Generation with CoT Reasoning and Layered Object Integration, Yuyao Zhang, Jinghao Li, Yu-Wing Tai, in NIPS 2025

**Questions:**

N/A

---

> ### Author Response · Authors · 2025-11-19
> **Reviewer xg6H— Response [1/2]**
>
> We sincerely thank the reviewer for the thoughtful and constructive comments. We address each point in detail in the following sections.
> We have also revised our paper according to the reviewers’ suggestions. In the updated manuscript, the revised content is highlighted in blue, and sections to be deleted or relocated are marked in green.
>
> **1. The main novelty is kind of limited; there are some similar works using CoT for image generation refinement, like LayerCraft [1]. Efficient attention is also a well-studied area. However, combining them might be the first try.**
>
> Thanks for highlighting this point. We agree that related works such as LayerCraft [1], as well as other CoT-based refinement frameworks and prior efficient-attention methods, were not thoroughly discussed in our initial submission. We have now revised the related-work section to provide a more comprehensive and precise discussion in main paper (page 9, lines 485–507).
>
> Regarding novelty, we would like to clarify our contributions along three dimensions:
> 1. **A novel unified cross-modal CoT framework within a single model.**
>
>    Our primary novelty lies in constructing a unified CoT model that jointly reasons over text and images within a *single* architecture. This enables *end-to-end* multimodal reasoning, where the model solves tasks across modalities without relying on external scaffolding or multi-agent coordination. In contrast, LayerCraft [1] adopts a multi-agent paradigm to orchestrate CoT for image generation. Although effective, its fragmented system limits generalization across tasks and prevents holistic optimization within a unified framework.
>
> 2. **A novel observation on the complexity bottleneck of multimodal reasoning.**
>
>    We also highlight a novel insight into why existing unified models struggle with interleaved reasoning. We identify that the main bottleneck arises from the substantial computational cost of processing visual tokens throughout the reasoning trajectory. This observation helps explain why, despite the release of multiple open-source unified models [2,3] over the past year, progress in achieving interleaved multimodal reasoning using a single model remains limited.
>
> 3. **A structural design that reduces complexity *while preserving reasoning ability*.**
>
>    While many efficient-attention mechanisms have been proposed for reducing the computational cost of LLMs, our design explicitly starts from the *requirement to maintain reasoning integrity*. Prior efficient-attention methods, as summarized in the survey [4], primarily focus on reducing attention complexity, often without considering the impact on multi-step reasoning behavior. Our design differs intrinsically in that it simultaneously targets complexity reduction and stable CoT-style reasoning, which are essential for multimodal decomposition.
>
> We hope these clarifications better articulate the conceptual and practical novelty of our approach beyond existing CoT-based image-generation or efficient-attention systems.
>
> [1] Zhang, Yuyao, Jinghao Li, and Yu-Wing Tai. "Layercraft: Enhancing text-to-image generation with cot reasoning and layered object integration." arXiv preprint arXiv:2504.00010 (2025).
>
> [2] Xie, Jinheng, et al. "Show-o: One single transformer to unify multimodal understanding and generation." arXiv preprint arXiv:2408.12528 (2024).
>
> [3] Chen, Xiaokang, et al. "Janus-pro: Unified multimodal understanding and generation with data and model scaling." arXiv preprint arXiv:2501.17811 (2025).
>
> [4] Sun, Yutao, et al. "Efficient attention mechanisms for large language models: A survey." arXiv preprint arXiv:2507.19595 (2025).

---

> ### Author Response · Authors · 2025-11-19
> **Reviewer xg6H— Response [2/2]**
>
> **2. The training dataset for CoT image generation purely depends on synthetic data. This limits generalization to open-domain or real-world visual reasoning tasks.**
>
> Thank you for pointing out this concern. The training dataset is dominated by synthetic with several real-world data (e.g., GeoPose3k [1]). We agree with reviewer's point that this limits generalization to open-domain or real-world visual reasoning tasks.
>
> To address this, we identify several real-world datasets that can be directly converted into multimodal CoT supervision for real-world tasks.
>
> - Embodied-AI datasets (e.g., Open X-Embodiment [2]) provide state–action trajectories that can be transformed into CoT traces, enabling the model to handle spatial reasoning tasks such as object manipulation and physical interaction.
>
> - Scientific instructional videos (e.g., JoVE [3]) can be segmented into step-wise visual–textual pairs, offering CoT supervision for causal and temporal tasks like multi-step procedures and action–outcome reasoning.
>
> - Non-fiction videos allow automatic extraction of causal event segments, which can be annotated with short rationales to support commonsense reasoning tasks in daily-life scenarios.
>
> By converting these datasets into structured CoT trajectories, our model is possible to handle real-world spatial, causal, temporal, and commonsense visual reasoning tasks. Due to the limited time and high complexity of these real-world tasks, we leave these in the future works.
>
> Additional detail of data collection is described in the revised Appendix D, page 21, line 1133-1164.
>
> [1] Brejcha, Jan, and Martin Čadík. "GeoPose3K: Mountain landscape dataset for camera pose estimation in outdoor environments." Image and Vision Computing 66 (2017): 1-14.
>
> [2] Vuong, Quan, et al. "Open x-embodiment: Robotic learning datasets and rt-x models." Towards Generalist Robots: Learning Paradigms for Scalable Skill Acquisition@ CoRL2023. 2023.
>
> [3] JoVE. JoVE: Journal of Visualized Experiments, Elsevier, https://www.jove.com. Accessed 19 Nov. 2025.
>
> **3. Figure S3 in the appendix seems not accurately reflect a realistic ground-level scene, which raises questions about the accuracy of synthetic training data.**
>
> Thanks for raising this concern. We agree that the initial visualization in Figure S3 does not fully resemble a realistic ground-level scene. Importantly, this issue is not due to inaccurate or synthetic data: the training samples are derived from real ground-level images in the GeoPose3k dataset [1] (page 23, line 1234). The limitation instead arises from two factors:
> (1) the GeoPose3k dataset is relatively small (3k training data),
> (2) the task involves a large viewpoint shift, making it intrinsically challenging，and
> (3) The ground-truth images contain a greater diversity of landscape types, but these are not directly linked to topographic or geographic features. As such, the model is prone to learn to disregard them during training, which in turn reduces the consistency with the ground truth in the generated outputs.
>
> We have updated Figure S3 (now Figure S4) in the appendix to provide a clearer and more representative comparison. While the generated images still show slight deviations, the revised figure demonstrates that Uni-CoT performs comparably to Nano-Banana (Google’s flagship model) on this highly challenging task, and significantly outperforms other large models such as Qwen-Image, Bagel, and GPT-5. This result further supports the effectiveness of our multimodal reasoning pipeline.
> Additionally, we have updated better visualization examples in Figure S7 (page 28). Notably, the new cases in Figure S7 highlight that our model has strong potential to generate ground-level scenes whose geometry closely align with the corresponding sohypse line map.
>
> [1] Brejcha, Jan, and Martin Čadík. "GeoPose3K: Mountain landscape dataset for camera pose estimation in outdoor environments." Image and Vision Computing 66 (2017): 1-14.

---

### Official Review · Reviewer_MA39 · 2025-11-02

**Soundness:** 3
**Presentation:** 3
**Contribution:** 3
**Rating:** 6
**Confidence:** 5

**Summary:**

This paper introduces Uni-CoT, a unified framework for multi-modal Chain-of-Thought (CoT) reasoning that integrates both textual and visual reasoning steps.  The proposed method employs a hierarchical macro-micro reasoning structure: the macro-level handles task decomposition and summarization, while the micro-level executes subtasks via a Markov Decision Process (MDP) with self-reflection.  The authors address the computational challenges of multi-modal CoT by reducing the complexity from quadratic to near-linear through localized reasoning and attention masking.  Experiments on image generation and understanding benchmarks (e.g., GenEval, WISE, MME, Jigsaw-R1) demonstrate state-of-the-art or competitive performance.

**Strengths:**

- The technical contribution under the unified model such as BAGEL is effective.
- The proposed method significantly reduces computational complexity, making long-horizon multi-modal reasoning more tractable.
- The paper includes both quantitative and qualitative results across multiple challenging benchmarks, including reasoning-driven generation and understanding tasks.
- Ablation on CoT mechanisms and training strategies provides strong evidence for the contribution of each component.
- The authors provide code, training details, and dataset construction pipelines, enhancing reproducibility.

**Weaknesses:**

- The evaluation results on understanding tasks are not sufficient. The authors only provide results on the MME benchmark, which is not representative of supporting the claim of unified tasks. Lack of more general and widely-used benchmarks such as MMMU, MMBench, OCRBench, MathVista, MathVision, etc.
- Experiments are conducted only on the BAGEL backbone.  It is unclear how the framework would perform with other MLLMs or in a more model-agnostic setting.

**Questions:**

Please refer to the weaknesses part. More results on understanding tasks will support the conclusions of this paper.

---

> ### Author Response · Authors · 2025-11-19
> **Reviewer MA39— Response**
>
> We sincerely thank the reviewer for the thoughtful and constructive comments. We address each point in detail in the following sections.
> We have also revised our paper according to the reviewers’ suggestions. In the updated manuscript, newly added content is highlighted in blue, and sections to be deleted or relocated are marked in green.
>
> **1. The evaluation results on understanding tasks are not sufficient.**
>
> Thanks for the evaluation concerns. We agree that a broader evaluation on understanding tasks would be helpful to support our claim on the performance of unified tasks. In our initial submission, we only conducted experiments on MME due to limited time and GPU resources. After the submission, we have refined our work to support a wider range of benchmarks by newly including the evaluation results of Uni-CoT and baseline methods on MMMU, MMBench and MathVista.
>
> As shown below, our model continues to slightly outperform the base model (BAGEL) on general understanding benchmarks such as MMMU, MMBench and MathVista, while maintaining its superior performance on reasoning-based multimodal benchmarks such as WISE and Jigsaw-R1. We have updated these results in Table 4 of the main paper (page 9, line 432-440).
>
>
> | Method | MME-P↑ | MME-S↑ | MMMU↑ | MMBench↑ | MathVista↑ | Jigsaw-R1↑ |
> |---|---:|---:|---:|---:|---:|---:|
> | Random | - | - | - | - | - | 40.63 |
> | GPT-4V | _1409_ | _1927_ | _56.8_ | _74.3_ | _47.5_ | - |
> | GPT-4.1-mini | - | - | - | - | - | _47.88_ |
> | InternVL2.5-2B | - | 2138 | 38.2 | - | 51.3 | 36.28 |
> | Qwen2.5-VL-7B | - | 2347 | **58.6** | 83.5 | 68.2 | 41.12 |
> | BAGEL | **1687** | 2388 | 52.8 | 85.0 | 73.1 | 40.73 |
> | Uni-CoT | **1687** | **2392** | 52.7 | **85.6** | **73.3** | **47.60** |

---

> ### Author Response · Authors · 2025-11-25
> **Update experimental results for Uni-CoT on extra backbone (BLIP3o)**
>
> **2. Experiments are conducted only on the BAGEL backbone. It is unclear how the framework would perform with other MLLMs or in a more model-agnostic setting.**
>
> Thank you for this valuable suggestion. We fully agree that evaluating Uni-CoT on additional unified multimodal models is important for demonstrating the generality of our framework. To address this, we extend Uni-CoT to **BLIP3o** [1], a recently proposed unified MLLM capable of both image understanding and image generation.
>
> More specifically, BLIP3o is built on an autoregressive Qwen2.5-VL backbone. For image understanding, it uses a CLIP encoder for image encoding and utilize autoregressive model to generate text answer. For image generation it first produces intermediate visual features via the autoregressive backbone, which are then passed to an external diffusion transformer to synthesize images.
>
> Here, we conduct experiments on WISE [2] benchmark and compare three variants to identify the effect of Uni-CoT’s micro-CoT (self-reflection):
>
> 1. **BLIP3o**: pretrained, no fine-tuning
> 2. **BLIP3o-SFT w/o Uni-CoT**: supervised fine-tuning without micro-CoT self-reflection
> 3. **BLIP3o-SFT w/ Uni-CoT**: supervised fine-tuningl with micro-CoT self-reflection
>
> As shown in below, directly fine-tuning BLIP3o already yields large gains, likely because the chosen BLIP3o variant (BLIP3o-NEXT-edit-VAE [3], we choose this variant for its competitive image editing performance) has relatively weak reasoning ability in its pretrained form.
> Importantly, **adding Uni-CoT further improves performance by ~0.8 points on average**, confirming that our reasoning architecture is model-agnostic and can consistently enhance multimodal reasoning quality.
>
> The corresponding reasoning trajectory visualizations are presented in the revised paper, page 30.
>
> | Model                      | Culture  | Time     | Space    | Biology  | Physics  | Chemistry | Overall ↑ |
> | -------------------------- | -------- | -------- | -------- | -------- | -------- | --------- | --------- |
> | **BLIP3o**                 | 0.30     | 0.33     | 0.48     | 0.37     | 0.36     | 0.22      | 0.33      |
> | **BLIP3o-SFT w/o Uni-CoT** | 0.54     | 0.54     | 0.59     | 0.42     | 0.61     | 0.57      | 0.55      |
> | **BLIP3o-SFT w/ Uni-CoT**  | **0.65** | **0.62** | **0.67** | **0.48** | **0.70** | **0.69**  | **0.64**  |
>
> [1] Chen, Jiuhai, et al. "Blip3-o: A family of fully open unified multimodal models-architecture, training and dataset." arXiv preprint arXiv:2505.09568 (2025).
>
> [2] Niu, Yuwei, et al. "Wise: A world knowledge-informed semantic evaluation for text-to-image generation." arXiv preprint arXiv:2503.07265 (2025).
>
> [3] Chen, Jiuhai, et al. BLIP3o/BLIP3o-NEXT-edit-VAE. Hugging Face, 2025. Available at: https://huggingface.co/BLIP3o/BLIP3o-NEXT-edit-VAE. Accessed 25 Nov 2025. License: Apache-2.0. Model: 5B params, BF16, Safetensors.

---

### Official Review · Reviewer_wVH1 · 2025-11-02

**Soundness:** 3
**Presentation:** 2
**Contribution:** 3
**Rating:** 6
**Confidence:** 2

**Summary:**

The paper proposes a unified multimodal Chain-of-Thought framework combining text and vision reasoning in one model. By introducing a macro–micro hierarchical structure and self-reflective MDP reasoning, it achieves coherent and efficient multimodal reasoning with reduced complexity. Results on multiple benchmarks show strong performance, though challenges remain in fine-grained visual consistency and generalization to dynamic scenes.

**Strengths:**

1. This paper proposes a unified framework that integrates textual and visual Chain-of-Thought reasoning in one model. It introduces a hierarchical macro–micro design that improves reasoning structure, coherence, and interpretability.
2. The method employs an MDP-based self-reflective process that reduces complexity from quadratic to near-linear while enhancing robustness.
3. The method achieves strong results on multiple benchmarks for both image generation and understanding.

**Weaknesses:**

1. Despite reduced complexity, multimodal reasoning remains computationally expensive due to the large number of visual tokens per step.
2. The dataset used for training is relatively small and partially synthetic, which may limit generalization to real-world multimodal scenarios.
3. The paper could be strengthened by providing a quantitative comparison of token usage during inference against baseline models.

**Questions:**

None

---

> ### Author Response · Authors · 2025-11-19
> **Reviewer wVH1 — Response [1/2]**
>
> We sincerely thank the reviewer for the thoughtful and constructive comments. We address each point in detail in the following sections. We have also revised our paper according to reviewers’ suggestions. In the updated manuscript, the revised content is highlighted in blue, and sections to be deleted or relocated are marked in green.
>
> **1. Challenges remain in fine-grained visual consistency and generalization to dynamic scenes.**
>
> We fully agree with the reviewer’s observation. As noted in the conclusion section, extending the framework to adapt to scenarios with strict requirements for visual understanding, which demand fine-grained visual consistency, remains an open direction to be explored. We are actively exploring several strategies to address this limitation. One promising direction is incorporating reinforcement learning to regularize image transitions and enforce consistency across reasoning steps. However, RL-based multimodal training remains computationally demanding and requires substantial GPU resources, making it challenging to complete at this stage. We consider this to be an important and meaningful direction for future work.
>
> **2. Despite reduced complexity, multimodal reasoning remains computationally expensive due to the large number of visual tokens per step.**
>
> Thanks for pointing this out. Although Uni-CoT reduces overall complexity, processing visual tokens at every reasoning step still incurs non-trivial cost. A potential improvement is to let the model adaptively decide when visual reasoning is necessary, reducing redundant visual-token usage. Additional complexity-reduction strategies include vision-token compression and performing CoT reasoning directly in latent space to save costs. We expect future work, ours or from the community, to explore these promising directions.
>
> **3. The dataset used for training is relatively small and partially synthetic, which may limit generalization to real-world multimodal scenarios.**
>
> Thank you for raising this important point. We agree that our current training set is relatively small and partially synthetic. However, our qualitative and quantative experiments demonstrate that the model can still learn multimodal reasoning abilities across several challenging benchmarks, proving that our dataset is effective and valuable.
>
> To further address concerns about *data scale* and *generalization*, we approach the problem from two directions:
>
> (1) The synthetic data pipeline is scalable.
>
> Our current synthetic CoT-generation pipeline can be conveniently expanded with modest additional compute/API resources. We estimate that an additional *300k high-quality multimodal CoT trajectories* can be produced with roughly *50K USD* and *500 GPU hours*, making large-scale synthetic augmentation feasible.
>
> (2) Building a scalable real-world multimodal dataset is feasible.
>
> We fully agree that real-world data is essential for improving generalization. Our investigation indicates that it is possible to obtain *~100k additional real multimodal CoT trajectories* from existing datasets with reasonable cost and effort. For example, Embodied-AI datasets (e.g., Open X-Embodiment [1]) contain large amounts of object-manipulation trajectories, which can strengthen real-world **spatial reasoning**. Scientific instructional videos such as JoVE [2] offer step-by-step procedural demonstrations, providing high-quality **causal** and **temporal** reasoning trajectories. And Non-fiction videos naturally contain diverse spatiotemporal events, providing abundant real-world **commonsense reasoning** data covering daily scenarios.
>
> Additional detail of data collection is described in the revised Appendix D, page 21, line 1133-1164.
>
> [1] Vuong, Quan, et al. "Open x-embodiment: Robotic learning datasets and rt-x models." Towards Generalist Robots: Learning Paradigms for Scalable Skill Acquisition@ CoRL2023. 2023.
>
> [2] JoVE. JoVE: Journal of Visualized Experiments, Elsevier, https://www.jove.com. Accessed 19 Nov. 2025.

---

> ### Author Response · Authors · 2025-11-19
> **Reviewer wVH1 — Response [2/2]**
>
> **4. The paper could be strengthened by providing a quantitative comparison of token usage during inference against baseline models.**
>
> We sincerely thank the reviewer for this important comment. We agree that a quantitative comparison of token usage during inference would better support our claims regarding simplified optimization and improved efficiency. In response, we conducted additional experiments to measure and compare the token usage incurred in interleaved reasoning trajectories between Uni-CoT and Uni-CoT Raw (Uni-CoT model without the proposed hierarchical framework).
>
> As shown in the table below, Uni-CoT Raw consumes substantially more tokens than Uni-CoT and incurs an almost quadratic increase in computation as the number of reasoning steps grows. In contrast, Uni-CoT requires substantial fewer tokens and exhibits near-linear scaling. Notably, with 2 and 3 multimodal reasoning steps, Uni-CoT reduces the average token interaction count by 2.24× and 2.95×, respectively, and the reduction grows to 11.26× when the number of reasoning steps reaches 10.
>
> | Reasoning Step Num | Uni-CoT  | Uni-CoT Raw |
> |--------------------|----------|-------------|
> | 0                  | 3.56e+07 | 8.45e+07    |
> | 1                  | 7.63e+07 | 1.71e+08    |
> | 2                  | 1.17e+08 | 3.46e+08    |
> | 3                  | 1.57e+08 | 6.11e+08    |
> | 4                  | 1.98e+08 | 9.68e+08    |
> | 5                  | 2.39e+08 | 1.41e+09    |
> | 6                  | 2.79e+08 | 1.95e+09    |
> | 7                  | 3.20e+08 | 2.57e+09    |
> | 8                  | 3.60e+08 | 3.28e+09    |
> | 9                  | 4.01e+08 | 4.09e+09    |
> | 10                 | 4.42e+08 | 4.98e+09    |
>
> These results and the corresponding analysis have been added to the main paper (page 9, line 470–481). We truly appreciate Reviewer wVH1’s insightful suggestion, which has helped strengthen the empirical evaluation of our work.
>
> Specifically, during the experiment, we calculate the average token interaction number incurred in the reasoning trajectories on 100 sampled prompt. We will re-organize the calculation code and release it through the anoymous github link since now it's a little messy, thanks again for this constructive suggestion.

---

### Author Response · Authors · 2025-12-01
**Paper Rebuttal Summarization for Area Chair**

Dear Area Chair,

We sincerely thank you for your time and careful coordination of the review process for our submission *“Uni-CoT: Towards Unified Chain-of-Thought Reasoning Across Text and Vision.”* We greatly appreciate the reviewers’ efforts and constructive feedback.

We understand that ACs need to invest substantial additional effort to ensure fair assessments in this time. To support your evaluation, we provide a concise summary of the reviewers’ key comments and our corresponding clarifications.

Across the evaluations, the paper received **four scores of 6 and one score of 4**, and reviewers consistently highlighted several notable strengths:

1. **Significant contribution to unified multi-modal reasoning** (wVH1, MA39, xg6H, XhV7).
   Reviewers emphasized that reasoning frameworks for unified models remain scarce, and Uni-CoT addresses this gap by enabling **genuine interleaved text–vision reasoning within a single model**, capability that current unified architectures have not yet robustly achieved.

2. **Effective hierachical architecture design** (wVH1, MA39, xg6H, XhV7).
   Our macro–micro CoT framework and MDP-based self-reflection were regarded as conceptually sound and practically effective, offering **a coherent, model-internal design** that improves reasoning structure, interpretability, and computational efficiency.

3. **Strong performance on challenging multimodal reasoning benchmarks** (wVH1, MA39, xg6H, XhV7).
   Uni-CoT achieves **state-of-the-art or competitive results** on WISE, GenEval, and Jigsaw-R1, producing intuitive and coherent reasoning trajectories across diverse reasoning-driven tasks.

4. **Meaningful computational advantages through principled complexity reduction** (wVH1, MA39, xg6H).
   Reviewers highlighted the practical significance of Uni-CoT’s localized-state formulation, which yields **near-linear multimodal CoT complexity** and effectively mitigates the dominant bottleneck of repeatedly processing large visual-token sequences.

5. **Clarity and reproducibility** (MA39, xg6H, XhV7).
   Reviewers praised the clarity of writing, intuitive figures, and comprehensive experiments. We also point out that the full pipeline and code have been open-sourced via an anonymous link:
   **[https://anonymous.4open.science/r/UniCoT/](https://anonymous.4open.science/r/UniCoT/)**.

In addition, several concerns were raised, for which we implemented corresponding revisions to strengthen the manuscript. We summarize them concisely below:

1. **Empirical support for the complexity-reduction claim** (wVH1, GGF1).
   *Revision:* We have conducted experiments to quantitatively compare Uni-CoT with a single-pass CoT (“Uni-CoT Raw”) on token-interaction comsumption. The results show a 2.2×–2.9× reduction for 2–3 steps and up to 11.3× reduction for longer trajectories (main paper, p.9, lines 468–481).

2. **Broader evaluation on multimodal understanding tasks** (MA39).
   *Revision:* We incorporated results on **MMMU, MMBench, and MathVista**, demonstrating that Uni-CoT maintains **stable and competitive understanding performance** (main paper, p.9, lines 432–440).

3. **Dataset scale and synthetic-data generalization** (wVH1, xg6H).
   *Revision:* We detailed a **scalable real-world multimodal CoT data strategy** (e.g., X-Embodiment, JoVE, scientific and commonsense video datasets). The real-data CoT extraction procedures are added in Appendic D (Supplementary Appendix, p.21, lines 1133-1164).

4. **Novelty concerns relative to prior scaffolding or hierarchical designs** (xg6H, XhV7, GGF1).
   *Revision:* We refined the novelty discussion by emphasizing three core contributions:
   - **A new observation that visual-token interaction is the primary bottleneck of unified multimodal reasoning.**
    - **A hierarchical CoT design explicitly constructed to reduce complexity while preserving multi-step reasoning fidelity.**
    - **A new MDP-based micro-CoT formulation**, tailored for multimodal self-reflection and tightly integrated with the hierarchical decomposition.

Notably,

   **Reviewer XhV7 indicated satisfaction with our clarification earlier.**

We hope this summary assists your evaluation of the paper's strengths, as well as our efforts addressing raised concerns during rebuttal. Thank you again for your time and consideration.

Sincerely,

The Authors

---

### Meta-Review · Area_Chair_TYHb · 2026-01-12

**Summary:**

- Multiple reviewers questioned whether the hierarchical CoT design constituted a significant advance over existing work in text-only CoT, hierarchical RL, multi-agent systems, or prompting-based scaffolding. There was a request for a clearer articulation of what is uniquely new.

- The understanding task evaluation was initially limited (only MME), lacking results on standard benchmarks like MMMU, MMBench, or MathVista.

- Experiments were conducted only on the BAGEL model, raising questions about the framework's generality.

- The claim of reducing computational complexity from quadratic to near-linear was seen as theoretical, lacking quantitative proof.

- The performance improvements over baselines on some benchmarks (e.g., WISE) were perceived as modest. The ablation studies did not clearly isolate the contributions of macro vs. micro components.

- Qualitative examples showed inconsistencies, indicating a remaining challenge in fine-grained visual alignment.

- Some reviewers initially found critical implementation details (base model, loss functions, training setup) omitted from the main paper, hindering assessment.

**Reviewer Concerns:**

Addressed Concerns:

- The authors provided a new quantitative experiment comparing token interaction counts between Uni-CoT and a "Uni-CoT Raw" baseline, showing near-linear vs. quadratic growth and concrete reduction factors (2.2x-2.9x for 2-3 steps).

- Broader Evaluation on Understanding Tasks. New results on MMMU, MMBench, and MathVista were added, showing stable, competitive performance.

- Generality Across Model Backbones. New experiments on BLIP3o were conducted, showing consistent performance gains from the Uni-CoT framework, supporting its model-agnostic claim.

- Clarity of Macro vs. Micro Contributions. The rebuttal and revised paper present ablation results showing macro-CoT is key for GenEval (prompt reorganization) and micro-CoT is key for WISE (fine-grained reasoning).

- Reproducibility & Missing Details. The authors state they have moved key implementation details from the appendix to the main paper and provided an anonymous code link.


Unaddressed Concerns:

- Core Novelty Perception. While the authors provided an argument for novelty, this is a matter of subjective judgment.

- Computational Expense. While complexity is reduced, the authors agree that processing visual tokens remains expensive and suggest adaptive mechanisms as future work.

**Reviewer Scores:**

All reviewers are likely to keep their scores.

---

### Decision · Program_Chairs · 2026-01-26

Accept (Poster)